# ROBUST REINFORCEMENT LEARNING FOR CONTINUOUS CONTROL WITH MODEL MISSPECIFICATION

**Daniel J. Mankowitz**[∗]**, Nir Levine**[∗]**, Rae Jeong, Abbas Abdolmaleki, Jost Tobias Springenberg, Yuanyuan Shi**[†]**, Jackie Kay, Timothy Mann, Todd Hester, Martin Riedmiller**
DeepMind
{dmankowitz, nirlevine, raejeong, aabdolmaleki, springenberg
yyshi, kayj, timothymann, toddhester, riedmiller}@google.com

## ABSTRACT

We provide a framework for incorporating robustness – to perturbations in the transition dynamics which we refer to as model misspecification – into continuous control Reinforcement Learning (RL) algorithms. We specifically focus on incorporating robustness into a state-of-the-art continuous control RL algorithm called Maximum a-posteriori Policy Optimization (MPO). We achieve this by learning a policy that optimizes for a worst case expected return objective and derive a corresponding robust entropy-regularized Bellman contraction operator. In addition, we introduce a less conservative, soft-robust, entropy-regularized objective with a corresponding Bellman operator. We show that both, robust and soft-robust policies, outperform their non-robust counterparts in nine Mujoco domains with environment perturbations. In addition, we show improved robust performance on a high-dimensional, simulated, dexterous robotic hand. Finally, we present multiple investigative experiments that provide a deeper insight into the robustness framework. This includes an adaptation to another continuous control RL algorithm as well as learning the uncertainty set from offline data. Performance videos can be found online at `https://sites.google.com/view/robust-rl`.

## 1 INTRODUCTION

Reinforcement Learning (RL) algorithms typically learn a *policy* that optimizes for the expected return (Sutton & Barto, 1998). That is, the policy aims to maximize the sum of future expected rewards that an agent accumulates in a particular task. This approach has yielded impressive results in recent years, including playing computer games with super human performance (Mnih et al., 2015; Tessler et al., 2016), multi-task RL (Rusu et al., 2016; Devin et al., 2017; Teh et al., 2017; Mankowitz et al., 2018b; Riedmiller et al., 2018) as well as solving complex continuous control robotic tasks (Duan et al., 2016; Abdolmaleki et al., 2018b; Kalashnikov et al., 2018; Haarnoja et al., 2018).

The current crop of RL agents are typically trained in a single environment (usually a simulator). As a consequence, an issue that is faced by many of these agents is the sensitivity of the agent's policy to environment perturbations. Perturbing the dynamics of the environment during test time, which may include executing the policy in a real-world setting, can have a significant *negative* impact on the performance of the agent (Andrychowicz et al., 2018; Peng et al., 2018; Derman et al., 2018; Di Castro et al., 2012; Mankowitz et al., 2018a). This is because the training environment is not necessarily a very good model of the perturbations that an agent may *actually* face, leading to potentially unwanted, sub-optimal behaviour. There are many types of environment perturbations. These include changing lighting/weather conditions, sensor noise, actuator noise, action delays etc (Dulac-Arnold et al., 2019).

It is desirable to train agents that are agnostic to environment perturbations. This is especially crucial in the Sim2Real setting (Andrychowicz et al., 2018; Peng et al., 2018; Wulfmeier et al., 2017; Rastogi et al., 2018; Christiano et al., 2016) where a policy is trained in a simulator and then executed on a

---

[∗]Equal contribution
[†]Work done during an internship at Deepmind

real-world domain. As an example, consider a robotic arm that executes a control policy to perform a specific task in a factory. If, for some reason, the arm needs to be replaced and the specifications do not exactly match, then the control policy still needs to be able to perform the task with the 'perturbed' robotic arm dynamics. In addition, sensor noise due to malfunctioning sensors, as well as actuator noise, may benefit from a robust policy to deal with these noise-induced perturbations.

**Model misspecification**: For the purpose of this paper, we refer to an agent that is trained in one environment and performs in a different, perturbed version of the environment (as in the above examples) as *model misspecification*. By incorporating robustness into our agents, we correct for this misspecification yielding improved performance in the perturbed environment(s).

In this paper, we propose a framework for incorporating robustness into continuous control RL algorithms. We specifically focus on robustness to model misspecification in the transition dynamics. Our main contributions are as follows:

**(1)** We incorporate robustness into a state-of-the-art continuous control RL algorithm called Maximum a-posteriori Policy Optimization (MPO) (Abdolmaleki et al., 2018b) to yield Robust MPO (R-MPO). We also carry out an additional experiment, where we incorporate robustness into an additional continuous RL algorithm called Stochastic Value Gradients (SVG) (Heess et al., 2015b).

**(2)** Entropy regularization encourages exploration and helps prevent early convergence to sub-optimal policies (Nachum et al., 2017). To incorporate these advantages, we: (i) Extend the Robust Bellman operator (Iyengar, 2005) to robust and soft-robust entropy-regularized versions, and show that these operators are contraction mappings. In addition, we (ii) extend MPO to *Robust* Entropy-regularized MPO (RE-MPO) and Soft RE-MPO (SRE-MPO) and show that they perform at least as well as R-MPO and in some cases significantly better. All the derivations have been deferred to Appendices B, C and D.

We want to emphasize that, while the theoretical contributions are novel, our most significant contribution is that of the extensive experimental analysis we have performed to analyze the robustness performance of our agent. Specifically:

**(3)** We present experimental results in nine Mujoco domains showing that RE-MPO, SRE-MPO and R-MPO, SR-MPO outperform both E-MPO and MPO respectively.

**(4)** To ensure that our method scales, we show robust performance on a high-dimensional, simulated, dexterous robotic hand called Shadow hand which outperforms the non-robust MPO baseline.

**(5)** Multiple investigative experiments to better understand the robustness framework. These include (i) an analysis of modifying the uncertainty set; (ii) comparing our technique to data augmentation; (iii) a comparison to domain randomization; (iv) comparing with and without entropy regularization; (v) We also train the transition models from offline data and use them as the uncertainty set to run R-MPO. We show that R-MPO with *learned* transition models as the uncertainty set can lead to improved performance over R-MPO.

## 2 BACKGROUND

**A Markov Decision Process (MDP)** is defined as the tuple $\langle S, A, r, \gamma, P \rangle$ where $S$ is the state space, $A$ the action space, $r : S \times A \to \mathbb{R}$ is a bounded reward function; $\gamma \in [0, 1]$ is the discount factor and $P : S \times A \to \Delta^S$ maps state-action pairs to a probability distribution over next states. We use $\Delta^S$ to denote the $|S| - 1$ simplex. The goal of a Reinforcement Learning agent for the purpose of control is to learn a policy $\pi : S \to \Delta^A$ which maps a state and action to a probability of executing the action from the given state so as to maximize the expected return $J(\pi) = \mathbb{E}^\pi[\sum_{t=0}^\infty \gamma^t r_t]$ where $r_t$ is a random variable representing the reward received at time $t$ (Sutton & Barto, 2018). The value function is defined as $V^\pi(s) = \mathbb{E}^\pi[\sum_{t=0}^\infty \gamma^t r_t | s_0 = s]$ and the action value function as $Q^\pi(s, a) = r(s, a) + \gamma \mathbb{E}_{s' \sim P(\cdot|s,a)}[V^\pi(s')]$.

**A Robust MDP (R-MDP)** is defined as a tuple $\langle S, A, r, \gamma, \mathcal{P} \rangle$ where $S, A, r$ and $\gamma$ are defined as above; $\mathcal{P}(s, a) \subseteq \mathcal{M}(S)$ is an uncertainty set where $\mathcal{M}(S)$ is the set of probability measures over next states $s' \in S$. This is interpreted as an agent selecting a state and action pair, and the next state $s'$ is determined by a conditional measure $p(s'|s, a) \in \mathcal{P}(s, a)$ (Iyengar, 2005). A robust policy optimizes for the worst-case expected return objective: $J_R(\pi) = \inf_{p \in \mathcal{P}} \mathbb{E}^{p,\pi}[\sum_{t=0}^\infty \gamma^t r_t]$.

The robust value function is defined as $V_R^\pi(s) = \inf_{p \in \mathcal{P}} \mathbb{E}^{p,\pi}[\sum_{t=0}^\infty \gamma^t r_t | s_0 = s]$ and the robust action value function as $Q_R^\pi(s,a) = r(s,a) + \gamma \inf_{p \in \mathcal{P}} \mathbb{E}_{s' \sim p(\cdot|s,a)}[V_R^\pi(s')]$. Both the robust Bellman operator $T_R^\pi : \mathcal{R}^{|S|} \to \mathcal{R}^{|S|}$ for a fixed policy and the optimal robust Bellman operator $T_R v(s) = \max_\pi T_R^\pi v(s)$ have previously been shown to be contractions (Iyengar, 2005). A rectangularity assumption on the uncertainty set (Iyengar, 2005) ensures that "nature" can choose a worst-case transition function independently for every state $s$ and action $a$.

**Maximum A-Posteriori Policy Optimization (MPO)** (Abdolmaleki et al., 2018a;b) is a continuous control RL algorithm that performs an expectation maximization form of policy iteration. There are two steps comprising **policy evaluation** and **policy improvement**. The *policy evaluation* step receives as input a policy $\pi_k$ and evaluates an action-value function $Q_\theta^{\pi_k}(s,a)$ by minimizing the squared TD error: $\min_\theta (r_t + \gamma Q_{\hat\theta}^{\pi_k}(s_{t+1} \sim P(\cdot|s_t, a_t), a_{t+1} \sim \pi_k(\cdot|s_{t+1})) - Q_\theta^{\pi_k}(s_t, a_t))^2$, where $\hat\theta$ denotes the parameters of a target network (Mnih et al., 2015) that are periodically updated from $\theta$. In practice we use a replay-buffer of samples in order to perform the policy evaluation step. The second step comprises a policy improvement step. The *policy improvement* step consists of optimizing the objective $\bar{J}(s,\pi) = \mathbb{E}_\pi[Q_\theta^{\pi_k}(s,a)]$ for states $s$ drawn from a state distribution $\mu(s)$. In practice the state distribution samples are drawn from an experience replay. By improving $\bar{J}$ in all states $s$, we improve our objective. To do so, a two step procedure is performed.

First, we construct a non-parametric estimate $q$ such that $\bar{J}(s, q) \geq \bar{J}(s, \pi_k)$. This is done by maximizing $\bar{J}(s, q)$ while ensuring that the solution, locally, stays close to the current policy $\pi_k$; i.e. $\mathbb{E}_{\mu(s)}[\mathrm{KL}(q(\cdot|s), \pi_k(\cdot|s))] < \epsilon$. This optimization has a closed form solution given as $q(a|s) \propto \pi_k(a|s) \exp^{Q_\theta^{\pi_k}(s,a)}/\eta$, where $\eta$ is a temperature parameter that can be computed by minimizing a convex dual function (Abdolmaleki et al. (2018b)). Second, we project this non-parametric representation back onto a parameterized policy by solving the optimization problem $\pi_{k+1} = \arg\min_\pi \mathbb{E}_{\mu(s)}[\mathrm{KL}(q(a|s)\|\pi(a|s)]$, where $\pi_{k+1}$ is the new and improved policy and where one typically employs additional regularization (Abdolmaleki et al., 2018a). Note that this amounts to supervised learning with samples drawn fron $q(a|s)$; see Abdolmaleki et al. (2018a) for details.

## 3 ROBUST MPO

To incorporate robustness into MPO, we focus on learning a worst-case value function in the policy evaluation step. Note that this policy evaluation step can be incorporated into any actor-critic algorithm. In particular, instead of optimizing the squared TD error, we optimize the worst-case squared TD error, which is defined as:

$$\min_\theta \left( r_t + \gamma \inf_{p \in \mathcal{P}(s_t, a_t)} \left[ Q_{\hat\theta}^{\pi_k}(s_{t+1} \sim p(\cdot|s_t, a_t), a_{t+1} \sim \pi_k(\cdot|s_{t+1})) \right] - Q_\theta^{\pi_k}(s_t, a_t) \right)^2, \quad (1)$$

where $\mathcal{P}(s_t, a_t)$ is an uncertainty set for the current state $s_t$ and action $a_t$; $\pi_k$ is the current network's policy, and $\hat\theta$ denotes the target network parameters. It is in this policy evaluation step (Line 3 in Algorithms 1,2 and 3 in Appendix I) that the Bellman operators in the previous sections are applied.

**Relation to MPO:** In MPO, this replaces the current policy evaluation step. The robust Bellman operator (Iyengar, 2005) ensures that this process converges to a unique fixed point for the policy $\pi_k$. This is achieved by repeated application of the robust Bellman operator during the policy evaluation step until convergence to the fixed point. Since the proposal policy $q(a|s)$ (see Section 2) is proportional to the robust action value estimate $Q_\theta^{\pi_k}(s,a)$, it intuitively yields a robust policy as the policy is being generated from a worst-case value function. The fitting of the policy network to the proposal policy yields a robust network policy $\pi_{k+1}$.

**Entropy-regularized MPO:** Entropy-regularization encourages exploration and helps prevent early convergence to sub-optimal policies (Nachum et al., 2017). To incorporate these advantages, we extended the Robust Bellman operator (Iyengar, 2005) to robust and soft-robust entropy-regularized versions (See Appendix B and C respectively for a detailed overview and the corresponding derivations) and show that these operators are contraction mappings (Theorem 1 below and Theorem 2 in Appendix E) and yield a well-known value-iteration bound with respect to the max norm.

**Theorem 1.** *The robust **entropy-regularized** Bellman operator $\mathcal{T}_{R\text{-}KL}^\pi$ for a fixed policy $\pi$ is a contraction operator. Specifically: $\forall U, V \in \mathbb{R}^{|S|}$ and $\gamma \in (0, 1)$, we have, $\|\mathcal{T}_{R\text{-}KL}^\pi U - \mathcal{T}_{R\text{-}KL}^\pi V\| \leq \gamma \|U - V\|$.*

In addition, we extended MPO to *Robust* Entropy-regularized MPO (RE-MPO) and Soft RE-MPO (SRE-MPO) (see Appendix D for a detailed overview and derivations) and show that they perform at least as well as R-MPO and in some cases significantly better. All the derivations have been deferred to the Appendix. The corresponding algorithms for R-MPO, RE-MPO and SRE-MPO can be found in Appendix I.

## 4 EXPERIMENTS

We now present experiments on nine different continuous control domains (four of which we show in the paper and the rest can be found in Appendix H.4) from the DeepMind control suite (Tassa et al., 2018). In addition, we present an experiment on a high-dimensional dexterous, robotic hand called Shadow hand (ShadowRobot, 2019). In our experiments, we found that the entropy-regularized version of Robust MPO had similar performance and in some cases, slightly better performance than the expected return version of Robust MPO *without* entropy-regularization. We therefore decided to include experiments of our agent optimizing the entropy-regularized objective (non-robust, robust and soft-robust versions). This corresponds to (a) non-robust E-MPO baseline, (b) Robust E-MPO (RE-MPO) and (c) Soft-Robust E-MPO (SRE-MPO). From hereon in, it is assumed that the algorithms optimize for the entropy-regularized objective unless otherwise stated.

*Appendix*: In Appendix H.4, we present results of our agent optimizing for the expected return objective *without* entropy regularization (for the non-robust, robust and soft-robust versions). This corresponds to (a') non-robust MPO baseline, (b') R-MPO and (c') SR-MPO.

The experiments are divided into three sections. The first section details the setup for robust and soft-robust training. The next section compares robust and soft-robust performance to the non-robust MPO baseline in each of the domains. The final section is a set of investigative experiments to gain additional insights into the performance of the robust and soft-robust agents.

**Setup:** For each domain, the robust agent is trained using a pre-defined uncertainty set consisting of three task perturbations [1]. Each of the three perturbations corresponds to a particular perturbation of the Mujoco domain. For example, in Cartpole, the uncertainty set consists of three different pole lengths. Both the robust and non-robust agents are evaluated on a test set of three unseen task perturbations. In the Cartpole example, this would correspond to pole lengths that the agent has not seen during training. The chosen values of the uncertainty set and evaluation set for each domain can be found in Appendix H.3. Note that it is common practice to manually select the pre-defined uncertainty set and the unseen test environments. Practitioners often have significant domain knowledge and can utilize this when choosing the uncertainty set (Derman & Mannor, 2019; Derman et al., 2018; Di Castro et al., 2012; Mankowitz et al., 2018a; Tamar et al., 2014).

During training, the robust, soft-robust and non-robust agents act in an unperturbed environment which we refer to as the *nominal* environment. During the TD learning update, the robust agent calculates an infimum between Q values from each next state realization for each of the uncertainty set task perturbations (the soft-robust agent computes an average, which corresponds to a uniform distribution over $\mathcal{P}$, instead of an infimum). Each transition model is a different instantiation of the Mujoco task. The robust and soft-robust agents are exposed to more state realizations than the non-robust agent. However, as we show in our ablation studies, significantly increasing the number of samples and the diversity of the samples for the non-robust agent still results in poor performance compared to the robust and soft-robust agents.

### 4.1 MAIN EXPERIMENTS

**Mujoco Domains:** We compare the performance of non-robust MPO to the robust and soft-robust variants. Each training run consists of $30k$ episodes and the experiments are repeated 5 times. In the bar plots, the y-axis indicates the average reward (with standard deviation) and the x-axis indicates different unseen evaluation environment perturbations starting from the first perturbation (Env0) onwards. Increasing environment indices correspond to increasingly large perturbations. For example, in Figure 1 (top left), Env0, Env1 and Env2 for the Cartpole Balance task represents the pole perturbed

---

[1]We did experiments on a larger set with similar results, but settled on three for computational efficiency.

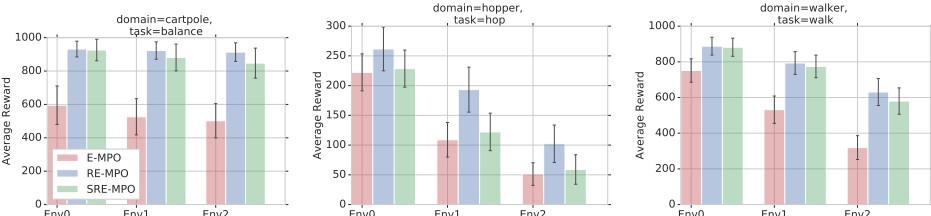

Figure 1: Three domains showing RE-MPO (blue), SRE-MPO (green) and E-MPO (red). The addition six domains can be found in the appendix. In addition, the results for R-MPO, SR-MPO and MPO can be found in Appendix H.4 with similar results.

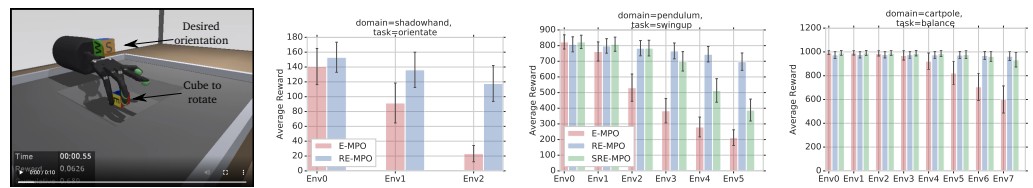

Figure 2: (1) The Shadow hand domain (left) and results for RE-MPO and E-MPO (middle left). (2) A larger test set: the figures show the performance of RE-MPO (blue), SRE-MPO (green) and E-MPO (red) for a test set that extends from the nominal environment to significant perturbations outside the training set for Cartpole Balance (middle right) and Pendulum Swingup (right).

to lengths of 2.0, 2.2 and 2.3 meters respectively. Figure 1 shows the performance of three Mujoco domains (The remaining six domains are in Appendix H.4). The bar plots indicate the performance of E-MPO (red), RE-MPO (blue) and SRE-MPO (green) on the held-out test perturbations. This color scheme is consistent throughtout the experiments unless otherwise stated. As can be seen in each of the figures, RE-MPO attains improved performance over E-MPO. This same trend holds true for all nine domains. SRE-MPO outperforms the non-robust baseline in all but the Cheetah domain, but is not able to outperform RE-MPO. An interesting observation can be seen in the video for the Walker walk task (https://sites.google.com/view/robust-rl), where the RE-MPO agent learns to 'drag' its leg which is a fundamentally different policy to that of the non-robust agent which learns a regular gait movement.

*Appendix:* The appendix contains additional experiments with the *non* entropy-regularized versions of the algorithms where again the robust (R-MPO) and soft robust (SR-MPO) versions of MPO outperform the non-robust version (MPO).

**Shadow hand domain:** This domain consists of a dexterous, simulated robotic hand called Shadow hand whose goal is to rotate a cube into a pre-defined orientation (ShadowRobot, 2019). The state space is a 79 dimensional vector and consisting of angular positions and velocities, the cube orientation and goal orientation. The action space is a 20 dimensional vector and consisting of the desired angular velocity of the hand actuators. The reward is a function of the current orientation of the cube relative to the desired orientation. The uncertainty set consists of three models which correspond to increasingly smaller sizes of the cube that the agent needs to orientate. The agent is evaluated on a different, unseen holdout set. The values can be found in Appendix H.3. We compare RE-MPO to E-MPO trained agents. Episodes are 200 steps long corresponding to approximately 10 seconds of interaction. Each experiment is run for $6k$ episodes and is repeated 5 times. As seen in Figure 2, RE-MPO outperforms E-MPO, especially as the size of the cube decreases (from Env0 to Env2). This is an especially challenging problem due to the high-dimensionality of the task. As seen in the videos (https://sites.google.com/view/robust-rl), the RE-MPO agent is able to manipulate significantly smaller cubes than it had observed in the nominal simulator.

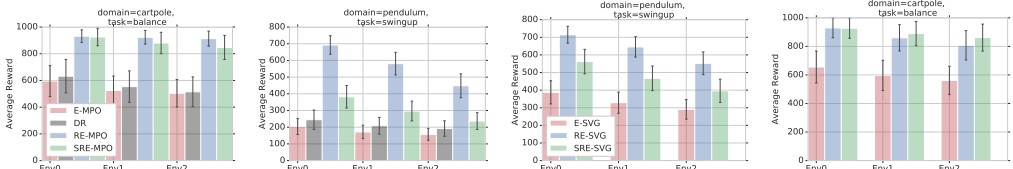

Figure 3: (1) Domain Randomization (DR): Domain randomization performance for the Cartpole balance (left) and Pendulum swingup (middle left) tasks. [3] (2) Stochastic Value Gradients (SVG): Two right images show the performance of Robust Entropy-regularized SVG (RE-SVG) and SRE-SVG compared to E-SVG for Pendulum and Cartpole respectively.

## 4.2 INVESTIGATIVE EXPERIMENTS

This section aims to investigate and try answer various questions that may aid in explaining the performance of the robust and non-robust agents respectively. Each investigative experiment is conducted on the Cartpole Balance and Pendulum Swingup domains.

**What if we increase the number of training samples?** One argument is that the robust agent has access to more samples since it calculates the Bellman update using the infimum of three different environment realizations. To balance this is effect, the non-robust agent was trained for three times more episodes than the robust agents. Training with significantly more samples does not increase the performance of the non-robust agent and, can even decreases the performance, as a result of overfitting to the nominal domain. See Appendix H.5, Figure 12 for the results.

**What about Domain Randomization?** A subsequent point would be that the robust agent sees more diverse examples compared to the non-robust agent from each of the perturbed environments. We therefore trained the non-robust agent in a domain randomization setting (Andrychowicz et al., 2018; Peng et al., 2018). We compare our method to two variants of DR. The first variant *Limited-DR* uses the same perturbations as in the uncertainty set of RE-MPO. Here, we compare which method better utilizes a limited set of perturbations to learn a robust policy. As seen in Figure 3 (left and middle left for Carpole Balance and Pendulum Swingup respectively), RE-MPO yields significantly better performance given the limited set of perturbations. The second variant *Full-DR* performs regular DR on a significantly larger set of 100 perturbations in the Pendulum Swingup task. In this setting, DR, which uses 30 times more perturbations, improves but still does not outperform RE-MPO (which still only uses three perturbations). This result can be seen in Figure 13, Appendix H.5.

**What is the intuitive difference between DR and RE-MPO/SRE-MPO? DR** defines the loss to be the expectation of TD-errors over the uncertainty set. Each TD error is computed using a state, action, reward, next state $< s, a, r, s' >$ trajectory from a particular perturbed environment, (selected uniformly from the uncertainty set). These TD errors are then averaged together. This is a form of data augmentation and the resulting behaviour is the average across all of this data. **RE-MPO/SRE-MPO**: In the case of robustness, the TD error is computed such that the *target* action value function is computed as a worst case value function with respect to the uncertainty set. This means that the learned policy is explicitly searching for adversarial examples during training to account for worst-case performance. In the soft-robust case, the subtle yet important difference (as seen in the experiments) with DR is that the TD loss is computed with the average *target* action value function with respect to next states (as opposed to averaging the TD errors of each individual perturbed environment as in DR). This results in different gradient updates being used to update the action value function compared to DR.

**A larger test set:** It is also useful to view the performance of the agent from the nominal environment to increasingly large perturbations in the unseen test set (see Appendix H.3 for values). These graphs can be seen in Figure 2 for Cartpole Balance and Pendulum Swingup respectively. As expected, the robust agent maintains a higher level of performance compared to the non-robust agent. Initially, the soft-robust agent outperforms the robust agent, but its performance degrades as the perturbations increase which is consistent with the results of Derman et al. (2018). In addition, the robust and soft-robust agents are competitive with the non-robust agent in the nominal environment.

Figure 4: Modifying the uncertainty set: Pendulum Swingup when modifying the third perturbation of the uncertainty set to values of 1.2 (left), 1.3 (middle) and 2.0 (right) meters respectively.

**Modifying the uncertainty set:** We now evaluate the performance of the agent for different uncertainty sets. For Pendulum Swingup, the original uncertainty set values of the pendulum arm are $1.0, 1.1$ and $1.4$ meters. We modified the final perturbation to values of $1.2, 1.3$ and $2.0$ meters respectively. The agent is evaluated on unseen lengths of $1.5, 1.6$ and $1.7$ meters. An increase in performance can be seen in Figure 4 as the third perturbation approaches that of the unseen evaluation environments. Thus it appears that if the agent is able to approximately capture the dynamics of the unseen test environments within the training set, then the robust agent is able to adapt to the unseen test environments. The results for cartpole balance can be seen in Appendix H.5, Figure 14.

**What about incorporating Robustness into other algorithms?** To show the generalization of this robustness approach, we incorporate it into the critic of the Stochastic Value Gradient (SVG) continuous control RL algorithm (See Appendix H.1). As seen in Figure 3, Robust Entropy-regularized SVG (RE-SVG) and Soft RE-SVG (SRE-SVG) significantly outperform the non-robust Entropy-regularized SVG (E-SVG) baseline in both Cartpole and Pendulum.

**Robust entropy-regularized return vs. robust expected return:** When comparing the robust entropy-regularized return performance to the robust expected return, we found that the entropy-regularized return appears to do no worse than the expected return. In some cases, e.g., Cheetah, the entropy-regularized objective performs significantly better (see Appendix H.5, Figure 11).

**Different Nominal Models:** In this paper the nominal model was always chosen as the smallest perturbation parameter value from the uncertainty set. This was done to highlight the strong performance of robust policies to increasingly large environment perturbations. However, what if we set the nominal model as the median or largest perturbation with respect to the chosen uncertainty set for each agent? As seen in Appendix H.5, Figure 15, the closer (further) the nominal model is to (from) the holdout set, the better (worse) the performance of the non-robust agent. However, in all cases, the robust agent still performs at least as well as (and sometimes better than) the non-robust agent.

**What about learning the uncertainty set from offline data?** In real-world settings, such as robotics and industrial control centers (Gao, 2014), there may be a nominal simulator available as well as offline data captured from the real-world system(s). These data could be used to train transition models to capture the dynamics of the task at hand. For example, a set of robots in a factory might each be performing the same task, such as picking up a box. In industrial control cooling centers, there are a number of cooling units in each center responsible for cooling the overall system. In both of these examples, each individual robot and cooling unit operate with slightly different dynamics due to slight fluctuations in the specifications of the designed system, wear-and-tear as well as sensor calibration errors. As a result, an uncertainty set of transition models can be trained from data generated by each robot or cooling unit.

However, can we train a set of transition models from these data, utilize them as the uncertainty set in R-MPO and still yield robust performance when training on a nominal simulator? To answer this question, we mimicked the above scenarios by generating datasets for the Cartpole Swingup and the Pendulum swingup tasks. For Cartpole swingup, we varied the length of the pole and generated a dataset for each pole length. For Pendulum Swingup, we varied the mass of the pole and generated the corresponding datasets. We then trained transition models on increasingly large data batches ranging from 100 to one million datapoints for each pole length and pole mass respectively. We then utilized each set of transition models for different data batch sizes as the uncertainty set and ran R-MPO on each task. We term this variant of R-MPO, Data-Driven Robust MPO (DDR-MPO). The results

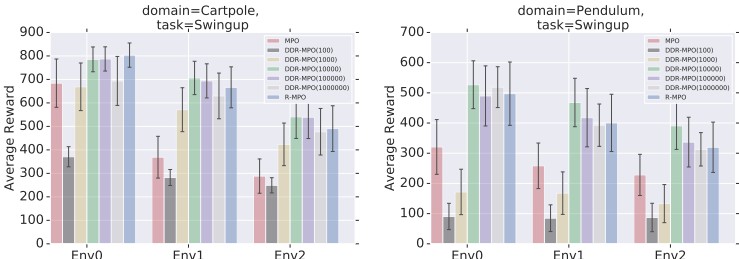

Figure 5: Training uncertainty sets of transition models on different batch sizes of offline data. The performance of Data Driven R-MPO (DDR-MPO) can be seen in the figures above for Cartpole Swingup (left) and Pendulum Swingup (right) respectively.

can be seen in Figure 5. There are a number of interesting observations from this analysis. (1) As expected, on small batches of data, the models are too inaccurate and result in poor performance. (2) An interesting insight is that as the data batch size increases, DDR-MPO starts to *outperform* R-MPO, especially for increasingly large perturbations. The hypothesis here is that due to the transition models being more accurate, but not perfect, adversarial examples are generated in a small region around the nominal next state observation, yielding an increasingly robust agent. (3) As the batch size increases further, and the transition models get increasingly close to the ground truth models, DDR-MPO converges to the performance of R-MPO.

## 5 RELATED WORK

From a theoretical perspective, Robust Bellman operators were introduced in (Iyengar, 2005; Nilim & Ghaoui, 2005; Wiesemann et al., 2013; Hansen & Sargent, 2011; Tamar et al., 2014). Our theoretical work extends this operator to the entropy regularized setting, for both the robust and soft-robust formulation, and modifies the MPO optimization formulation accordingly. A more closely related work from a theoretical perspective is that of Grau-Moya et al. (2016) who introduces a formulation for robustness to model misspecification. This work is a special case of robust MDPs where they introduce a robust Bellman operator that regularizes the *immediate* reward with two KL terms; one entropy term capturing model uncertainty with respect to a base model, and the other term being entropy regularization with respect to a base policy. Our work differs from this work in a number of respects: (1) Their uncertainty set is represented by a KL constraint which has the effect of restricting the set of admissible transition models. Our setup does not have these restrictions. (2) The uncertainty set elements from Grau-Moya et al. (2016) output a probability distribution over model parameter space whereas the uncertainty set elements in our formulation output a distribution over next states.

Mankowitz et al. (2018a) learn robust options, also known as temporally extended actions (Sutton et al., 1999), using policy gradient. Robust solutions tend to be overly conservative. To combat this, Derman et al. (2018) extend the actor-critic two-timescale stochastic approximation algorithm to a 'soft-robust' formulation to yield a less, conservative solution. Di Castro et al. (2012) introduce a robust implementation of Deep Q Networks (Mnih et al., 2015). Domain Randomization (DR) (Andrychowicz et al., 2018; Peng et al., 2018) is a technique whereby an agent trains on different perturbations of the environment. The agent batch averages the learning error of these different perturbed trajectories together to yield an agent that is robust to environment perturbations. This can be viewed as a data augmentation technique where the resulting behaviour is the average across all of the data. There are also works that look into robustness to action stochasticity (Fox et al., 2015; Braun et al., 2011; Rubin et al., 2012).

## 6 CONCLUSION

We have presented a framework for incorporating robustness - to perturbations in the transition dynamics, which we refer to as model misspecification - into continuous control RL algorithms. This framework is suited to continuous control algorithms that learn a value function, such as an actor critic setup. We specifically focused on incorporating robustness into MPO as well as our entropy-regularized version of MPO (E-MPO). In addition, we presented an experiment which incorporates robustness into the SVG algorithm. From a *theoretical* standpoint, we adapted MPO to an entropy-

regularized version (E-MPO); we then incorporated robustness into the policy evaluation step of both algorithms to yield Robust MPO (R-MPO) and Robust E-MPO (RE-MPO) as well as the soft-robust variants (SR-MPO/SRE-MPO). This was achieved by deriving the corresponding robust and soft-robust entropy-regularized Bellman operators to ensure that the policy evaluation step converges in each case. We have extensive experiments showing that the robust versions outperform the non-robust counterparts on nine Mujoco domains as well as a high-dimensional dexterous, simulated robotic hand called Shadow hand (ShadowRobot, 2019). We also provide numerous investigative experiments to understand the robust and soft-robust policy in more detail. This includes an experiment showing improved robust performance over R-MPO when using an uncertainty set of transition models learned from offline data.

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

# A  BACKGROUND

**Entropy-regularized Reinforcement Learning**: Entropy regularization encourages exploration and helps prevent early convergence to sub-optimal policies (Nachum et al., 2017). We make use of the relative entropy-regularized RL objective defined as $J_{\text{KL}}(\pi; \bar{\pi}) = \mathbb{E}^\pi[\sum_{t=0}^\infty \gamma^t(r_t - \tau\text{KL}(\pi(\cdot|s_t)\|\bar{\pi}(\cdot|s_t)))]$ where $\tau$ is a temperature parameter and $\text{KL}(\pi(\cdot|s_t)\|\bar{\pi}(\cdot|s_t))$ is the Kullback-Leibler (KL) divergence between the current policy $\pi$ and a reference policy $\bar{\pi}$ given a state $s_t$ (Schulman et al., 2017). The entropy-regularized value function is defined as $V_{\text{KL}}^\pi(s; \bar{\pi}) = \mathbb{E}^\pi[\sum_{t=0}^\infty \gamma^t(r_t - \tau\text{KL}(\pi(\cdot|s_t)\|\bar{\pi}(\cdot|s_t)))|s_0 = s]$. Intuitively, augmenting the rewards with the KL term regularizes the policy by forcing it to be 'close' in some sense to the base policy.

# B  ROBUST ENTROPY-REGULARIZED BELLMAN OPERATOR

(Relative-)Entropy regularization has been shown to encourage exploration and prevent early convergence to sub-optimal policies (Nachum et al., 2017). To take advantage of this idea when developing a robust RL algorithm we extend the robust Bellman operator to a robust *entropy regularized* Bellman operator and prove that it is a contraction.[4] We also show that well-known value iteration bounds can be attained using this operator. We first define the **robust** entropy-regularized value function as $V_{\text{R-KL}}^\pi(s; \bar{\pi}) = \mathbb{E}_{a\sim\pi(\cdot|s)}[r(s,a) - \tau\log\frac{\pi(\cdot|s)}{\bar{\pi}(\cdot|s)} + \gamma\inf_{p\in\mathcal{P}}\mathbb{E}_{s'\sim p(\cdot|s,a)}[V_{\text{R-KL}}^\pi(s'; \bar{\pi})]]$. For the remainder of this section, we drop the sub-and superscripts, as well as the reference policy conditioning, from the value function $V_{\text{R-KL}}^\pi(s; \bar{\pi})$, and simply represent it as $V(s)$ for brevity. We define the robust entropy-regularized Bellman operator for a fixed policy $\pi$ in Equation 2, and show it is a max norm contraction (Theorem 1).

$$\mathcal{T}_{\text{R-KL}}^\pi V(s) = \mathbb{E}_{a\sim\pi(\cdot|s)}[r(s,a) - \tau\log\frac{\pi(\cdot|s)}{\bar{\pi}(\cdot|s)} + \gamma\inf_{p\in\mathcal{P}}\mathbb{E}_{s'\sim p(\cdot|s,a)}[V(s')]] \ , \qquad (2)$$

**Theorem 1.** *The robust **entropy-regularized** Bellman operator $\mathcal{T}_{R\text{-}KL}^\pi$ for a fixed policy $\pi$ is a contraction operator. Specifically: $\forall U, V \in \mathbb{R}^{|S|}$ and $\gamma \in (0,1)$, we have, $\|\mathcal{T}_{R\text{-}KL}^\pi U - \mathcal{T}_{R\text{-}KL}^\pi V\| \leq \gamma\|U - V\|$.*

The proof can be found in the (Appendix E, Theorem 1). Using the optimal robust entropy-regularized Bellman operator $T_{\text{R-KL}} = \sup_\pi T_{\text{R-KL}}^\pi$, which is shown to also be a contraction operator in Appendix E, Theorem 2, a standard value iteration error bound can be derived (Appendix E, Corollary 1).

# C  *Soft*-ROBUST ENTROPY-REGULARIZED BELLMAN OPERATOR

In this section, we derive a soft-robust entropy-regularized Bellman operator and show that it is also a $\gamma$-contraction in the max norm. First, we define the average transition model as $\bar{p} = \mathbb{E}^{p\sim w}[p]$ which corresponds to the average transition model distributed according to some distribution $w$ over the uncertainty set $\mathcal{P}$. This average transition model induces an average stationary distribution (see Derman et al. (2018)). The *soft-robust* entropy-regularized value function is defined as $V_{\text{SR-KL}}^\pi(s; \bar{\pi}) = \mathbb{E}_{a\sim\pi(\cdot|s)}[r(s,a) - \tau\log\frac{\pi(\cdot|s)}{\bar{\pi}(\cdot|s)}] + \gamma\mathbb{E}_{s'\sim\bar{p}(\cdot|s,a)}[V_{\text{SR-KL}}^\pi(s'; \bar{\pi})]$. Again, for ease of notation, we denote $V_{\text{SR-KL}}^\pi(s; \bar{\pi}) = V(s)$ for the remainder of the section. The soft-robust entropy-regularized Bellman operator for a fixed policy $\pi$ is defined as:

$$\mathcal{T}_{\text{SR-KL}}^\pi V(s) = \mathbb{E}_{a\sim\pi(\cdot|s)}[r(s,a) - \tau\log\frac{\pi(\cdot|s)}{\bar{\pi}(\cdot|s)} + \gamma\mathbb{E}_{s'\sim\bar{p}(\cdot|s,a)}[V(s')]] \ , \qquad (3)$$

which is also a contraction mapping (see Appendix F, Theorem 3) and yields the same bound as Corollary 1 for the optimal soft-robust Bellman operator derived in Appendix F, Theorem 4.

# D  ROBUST ENTROPY-REGULARIZED POLICY EVALUATION

To extend Robust policy evaluation to robust *entropy-regularized* policy evaluation, two key steps need to be performed: (1) optimize for the entropy-regularized expected return as opposed to the regular expected return and modify the TD update accordingly; (2) Incorporate

---

[4]Note that while MPO already bounds the per step relative entropy we, in addition, want to regularize the action-value function to obtain a robust regularized algorithm.

robustness into the entropy-regularized expected return and modify the entropy-regularized TD update. To achieve (1), we define the entropy-regularized expected return as $Q_{\text{KL}}^{\bar{\pi}_k}(s, a; \bar{\pi}) = r(s, a) - \tau \text{KL}(\pi_k(\cdot|s)\|\bar{\pi}(\cdot|s)) + \mathbb{E}_{s' \sim p(\cdot|s,a)}[V_{\text{KL}}^{\pi_k}(s'; \bar{\pi})]$, and show in Appendix G that performing policy evaluation with the entropy-regularized value function is equivalent to optimizing the entropy-regularized squared TD error (same as Eq. equation 4, only omitting the inf operator). To achieve (2), we optimize for the robust entropy regularized expected return objective defined as $Q_{\text{R-KL}}^{\pi_k}(s, a; \bar{\pi}) = r(s, a) - \tau \text{KL}(\pi_k(\cdot|s)\|\bar{\pi}(\cdot|s)) + \inf_{p \in \mathcal{P}} \mathbb{E}_{s' \sim p(\cdot|s,a)}[V_{\text{R-KL}}^{\pi_k}(s'; \bar{\pi})]$, yielding the robust entropy-regularized squared TD error:

$$
\min_\theta \Bigg( r_t + \gamma \inf_{p \in \mathcal{P}(s_t, a_t)} \bigg[ \widetilde{Q}_{\text{R-KL}, \hat{\theta}}^{\pi_k}(s_{t+1} \sim p(\cdot|s_t, a_t), a_{t+1} \sim \pi_k(\cdot|s_{t+1}); \bar{\pi})
$$

$$
- \tau \text{KL}(\pi_k(\cdot|s_{t+1} \sim p(\cdot|s_t, a_t))\|\bar{\pi}(\cdot|s_{t+1} \sim p(\cdot|s_t, a_t))) \bigg] - \widetilde{Q}_{\text{R-KL}, \theta}^{\pi_k}(s_t, a_t; \bar{\pi}) \Bigg)^2, \tag{4}
$$

where $Q_{\text{R-KL}}^{\pi_k}(s, a; \bar{\pi}) = \widetilde{Q}_{\text{R-KL}}^{\pi_k}(s, a; \bar{\pi}) - \tau \text{KL}(\pi_k(\cdot|s)\|\bar{\pi}(\cdot|s))$. For the *soft-robust* setting, we remove the infimum from the TD update and replace the next state transition function $p(\cdot|s_t, a_t)$ with the average next state transition function $\bar{p}(\cdot|s_t, a_t)$.

**Relation to MPO:** As in the previous section, this step replaces the policy evaluation step of MPO. Our robust *entropy-regularized* Bellman operator $T_{\text{R-KL}}^{\pi_k}$ and soft-robust *entropy-regularized* Bellman operator $T_{\text{SR-KL}}^{\pi_k}$ ensures that this process converges to a unique fixed point for the policy $\pi_k$ for the robust and soft-robust cases respectively. We use $\pi_{k-1}$ as the reference policy $\bar{\pi}$. The pseudo code for the R-MPO, RE-MPO and Soft-Robust Entropy-regularized MPO (SRE-MPO) algorithms can be found in Appendix I (Algorithms 1, 2 and 3 respectively).

# E PROOFS

**Theorem 1.**

*Proof.* We follow the proofs from (Tamar et al., 2014; Iyengar, 2005), and adapt them to account for the additional entropy regularization for a fixed policy $\pi$. Let $U, V \in \mathbb{R}^{|S|}$, and $s \in S$ an arbitrary state. Assume $\mathcal{T}_{\text{R-KL}}^\pi U(s) \geq \mathcal{T}_{\text{R-KL}}^\pi V(s)$. Let $\epsilon > 0$ be an arbitrary positive number.

By the definition of the inf operator, there exists $p_s \in \mathcal{P}$ such that,

$$
\mathbb{E}_{a \sim \pi(\cdot|s)}[r(s, a) - \tau \log \frac{\pi(\cdot|s)}{\bar{\pi}(\cdot|s)} + \gamma \mathbb{E}_{s' \sim p_s(\cdot|s,a)}[V(s')]]
$$

$$
< \inf_{p \in \mathcal{P}} \mathbb{E}_{a \sim \pi(\cdot|s)}[r(s, a) - \tau \log \frac{\pi(\cdot|s)}{\bar{\pi}(\cdot|s)} + \gamma \mathbb{E}_{s' \sim p(\cdot|s,a)}[V(s')]] + \epsilon \tag{5}
$$

In addition, we have by definition that:

$$
\mathbb{E}_{a \sim \pi(\cdot|s)}[r(s, a) - \tau \log \frac{\pi(\cdot|s)}{\bar{\pi}(\cdot|s)} + \gamma \mathbb{E}_{s' \sim p_s(\cdot|s,a)}[U(s')]]
$$

$$
\geq \inf_{p \in \mathcal{P}} \mathbb{E}_{a \sim \pi(\cdot|s)}[r(s, a) - \tau \log \frac{\pi(\cdot|s)}{\bar{\pi}(\cdot|s)} + \gamma \mathbb{E}_{s' \sim p(\cdot|s,a)}[U(s')]] \tag{6}
$$

Thus, we have,

$$0 \leq \mathcal{T}_{\text{R-KL}}^{\pi} U(s) - \mathcal{T}_{\text{R-KL}}^{\pi} V(s)$$

$$< \mathbb{E}_{a \sim \pi(\cdot|s)} \left[ r(s,a) - \tau \log \frac{\pi(\cdot|s)}{\bar{\pi}(\cdot|s)} + \gamma \mathbb{E}_{s' \sim p_s(\cdot|s,a)} [U(s')] \right]$$

$$- \mathbb{E}_{a \sim \pi(\cdot|s)} \left[ r(s,a) - \tau \log \frac{\pi(\cdot|s)}{\bar{\pi}(\cdot|s)} + \gamma \mathbb{E}_{s' \sim p_s(\cdot|s,a)} [V(s')] \right] + \epsilon \qquad (7)$$

$$= \mathbb{E}_{a \sim \pi(\cdot|s), s' \sim p_s(\cdot|s,a)} [\gamma U(s')] - \mathbb{E}_{a \sim \pi(\cdot|s), s' \sim p_s(\cdot|s,a)} [\gamma V(s')] + \epsilon$$

$$\leq \gamma \|U - V\| + \epsilon$$

Applying a similar argument for the case $\mathcal{T}_{\text{R-KL}} U(s) \leq \mathcal{T}_{\text{R-KL}} V(s)$ results in

$$|\mathcal{T}_{\text{R-KL}}^{\pi} U - \mathcal{T}_{\text{R-KL}}^{\pi} V| < \gamma \|U - V\| + \epsilon. \qquad (8)$$

Since $\epsilon$ is an arbitrary positive number, we establish the result, i.e.,

$$|\mathcal{T}_{\text{R-KL}}^{\pi} U - \mathcal{T}_{\text{R-KL}}^{\pi} V| \leq \gamma \|U - V\|. \qquad (9)$$

$$\square$$

**Theorem 2.**

*Proof.* We follow a similar argument to the proof of Theorem 1. Let $U, V \in \mathbb{R}^{|S|}$, and $s \in S$ an arbitrary state. Assume $\mathcal{T}_{\text{R-KL}}U(s) \geq \mathcal{T}_{\text{R-KL}}V(s)$. Let $\epsilon > 0$ be an arbitrary positive number. By definition of the sup operator, there exists $\hat{\pi} \in \Pi$ such that,

$$\inf_{p \in \mathcal{P}} \mathbb{E}_{a \sim \hat{\pi}(\cdot|s)}[r(s,a) - \tau \log \frac{\hat{\pi}(\cdot|s)}{\bar{\pi}(\cdot|s)} + \gamma \mathbb{E}_{s' \sim p(\cdot|s,a)}[U(s')]] > \mathcal{T}_{\text{R-KL}}U(s) - \epsilon \qquad (10)$$

In addition, by the definition of the inf operator, there exists $p_s \in \mathcal{P}$ such that,

$$\mathbb{E}_{a \sim \hat{\pi}(\cdot|s)}[r(s,a) - \tau \log \frac{\hat{\pi}(\cdot|s)}{\bar{\pi}(\cdot|s)} + \gamma \mathbb{E}_{s' \sim p_s(\cdot|s,a)}[V(s')]]$$

$$< \inf_{p \in \mathcal{P}} \mathbb{E}_{a \sim \hat{\pi}(\cdot|s)}[r(s,a) - \tau \log \frac{\hat{\pi}(\cdot|s)}{\bar{\pi}(\cdot|s)} + \gamma \mathbb{E}_{s' \sim p(\cdot|s,a)}[V(s')]] + \epsilon \qquad (11)$$

Thus, we have,

$$0 \leq \mathcal{T}_{\text{R-KL}}U(s) - \mathcal{T}_{\text{R-KL}}V(s)$$

$$< (\inf_{p \in \mathcal{P}} \mathbb{E}_{a \sim \hat{\pi}(\cdot|s)}[r(s,a) - \tau \log \frac{\hat{\pi}(\cdot|s)}{\bar{\pi}(\cdot|s)} + \gamma \mathbb{E}_{s' \sim p(\cdot|s,a)}[U(s')]] + \epsilon)$$

$$- (\inf_{p \in \mathcal{P}} \mathbb{E}_{a \sim \hat{\pi}(\cdot|s)}[r(s,a) - \tau \log \frac{\hat{\pi}(\cdot|s)}{\bar{\pi}(\cdot|s)} + \gamma \mathbb{E}_{s' \sim p(\cdot|s,\hat{\pi}(a|s))}[V(s')]])$$

$$< (\mathbb{E}_{a \sim \hat{\pi}(\cdot|s)}[r(s,a) - \tau \log \frac{\hat{\pi}(\cdot|s)}{\bar{\pi}(\cdot|s)} + \gamma \mathbb{E}_{s' \sim p_s(\cdot|s,a)}[U(s')]] + \epsilon) \qquad (12)$$

$$- (\mathbb{E}_{a \sim \hat{\pi}(\cdot|s)}[r(s,a) - \tau \log \frac{\hat{\pi}(\cdot|s)}{\bar{\pi}(\cdot|s)} + \gamma \mathbb{E}_{s' \sim p_s(\cdot|s,a)}[V(s')]] - \epsilon)$$

$$= \mathbb{E}_{a \sim \hat{\pi}(\cdot|s), s' \sim \bar{p}(\cdot|s,a)}[\gamma U(s')] - \mathbb{E}_{a \sim \hat{\pi}(\cdot|s), s' \sim \bar{p}(\cdot|s,a)}[\gamma V(s')] + 2\epsilon$$

$$\leq \gamma \|U - V\| + 2\epsilon$$

Applying a similar argument for the case $\mathcal{T}_{\text{R-KL}}U(s) \leq \mathcal{T}_{\text{R-KL}}V(s)$ results in

$$|\mathcal{T}_{\text{R-KL}}U - \mathcal{T}_{\text{R-KL}}V| < \gamma \|U - V\| + 2\epsilon. \qquad (13)$$

Since $\epsilon$ is an arbitrary positive number, we establish the result, i.e.,

$$|\mathcal{T}_{\text{R-KL}}U - \mathcal{T}_{\text{R-KL}}V| \leq \gamma \|U - V\|. \qquad (14)$$

$$\square$$

**Corollary 1.** *Let $\pi_N$ be the greedy policy after applying $N$ value iteration steps. The bound between the optimal value function $V^*$ and $V^{\pi_N}$, the value function that is induced by $\pi_N$, is given by, $\|V^* - V^{\pi_N}\| \leq \frac{2\gamma\epsilon}{(1-\gamma)^2} + \frac{2\gamma^{N+1}}{(1-\gamma)}\|V^* - V_0\|$, where $\epsilon = \max_{0 \leq k \leq N} \|\mathcal{T}_{R\text{-}KL}V_k - V_{k+1}\|$ is the function approximation error, and $V_0$ is the initial value function.*

*Proof.* From Berteskas (1996), we have the following proposition:

Let $V^*$ be the optimal value function, $V$ some arbitrary value function, $\pi$ the greedy policy with respect to $V$, and $V^\pi$ the value function that is induced by $\pi$. Thus,

$$\|V^* - V^\pi\| \leq \frac{2\gamma}{(1-\gamma)}\|V^* - V\| \tag{15}$$

Next, define the maximum projected loss to be:

$$\epsilon = \max_{0 \leq k \leq N} \|\mathcal{T}_{\text{R-KL}}V_k - V_{k+1}\| \tag{16}$$

We can now derive a bound on the loss between the optimal value function $V^*$ and the value function obtained after $N$ updates of value iteration (denoted by $V_N$) as follows:

$$\begin{aligned}
\|V^* - V_N\| &\leq \|V^* - \mathcal{T}_{\text{R-KL}}V_{N-1}\| + \|\mathcal{T}_{\text{R-KL}}V_{N-1} - V_N\| \\
&= \|\mathcal{T}_{\text{R-KL}}V^* - \mathcal{T}_{\text{R-KL}}V_{N-1}\| + \|\mathcal{T}_{\text{R-KL}}V_{N-1} - V_N\| \\
&\leq \gamma\|V^* - V_{N-1}\| + \|\mathcal{T}_{\text{R-KL}}V_{N-1} - V_N\| \\
&\leq \gamma\|V^* - V_{N-1}\| + \epsilon \\
&\leq (1 + \gamma + \cdots + \gamma^{N-1})\epsilon + \gamma^N\|V^* - V_0\| \\
&\leq \frac{\epsilon}{(1-\gamma)} + \gamma^N\|V^* - V_0\|
\end{aligned} \tag{17}$$

Then, using Lemma E, we get:

$$\begin{aligned}
\|V^* - V^{\pi_N}\| &\leq \frac{2\gamma}{(1-\gamma)}\|V^* - V_N\| \\
&\leq \frac{2\gamma}{(1-\gamma)}\frac{\epsilon}{(1-\gamma)} + \frac{2\gamma}{(1-\gamma)}\gamma^N\|V^* - V_0\| \\
&= \frac{2\gamma\epsilon}{(1-\gamma)^2} + \frac{2\gamma^{N+1}}{(1-\gamma)}\|V^* - V_0\|
\end{aligned} \tag{18}$$

which establishes the result. $\qquad\square$

## F  SOFT-ROBUST ENTROPY-REGULARIZED BELLMAN OPERATOR

**Theorem 3.**

*Proof.* For an arbitrary $U, V \in \mathbb{R}^{|S|}$ and for a fixed policy $\pi$:

$$\|\mathcal{T}^\pi_{\text{SR-KL}}U(s) - \mathcal{T}^\pi_{\text{SR-KL}}V(s)\|_\infty$$

$$= \sup_s \left| \mathbb{E}_{a \sim \pi(\cdot|s)}[r(s,a) - \tau \log \frac{\pi(\cdot|s)}{\bar{\pi}(\cdot|s)} + \gamma \mathbb{E}_{s' \sim \bar{p}(\cdot|s,a)}[U(s')]] \right.$$

$$\left. - \mathbb{E}_{a \sim \pi(\cdot|s)}[r(s,a) - \tau \log \frac{\pi(\cdot|s)}{\bar{\pi}(\cdot|s)} + \gamma \mathbb{E}_{s' \sim \bar{p}(\cdot|s,a)}[V(s')]] \right|$$

$$= \gamma \sup_s |\sum_{s'} \bar{p}(s'|s,a)[U(s') - V(s')]|$$

$$\leq \gamma \sup_s \sum_{s'} \bar{p}(s'|s,a)|U(s') - V(s')|$$

$$\leq \gamma \sup_s \sum_{s'} \bar{p}(s'|s,a)\|U(s') - V(s')\|_\infty$$

$$\leq \gamma \|U - V\|_\infty$$

$\square$

**Theorem 4.**

*Proof.* Let $U, V \in \mathbb{R}^{|S|}$, and $s \in S$ an arbitrary state. Assume $\mathcal{T}_{\text{SR-KL}}U(s) \geq \mathcal{T}_{\text{SR-KL}}V(s)$. Let $\epsilon > 0$ be an arbitrary positive number. By definition of the $\sup$ operator, there exists $\hat{\pi} \in \Pi$ such that,

$$\mathbb{E}_{a \sim \hat{\pi}(\cdot|s)}[r(s,a) - \tau \log \frac{\hat{\pi}(\cdot|s)}{\bar{\pi}(\cdot|s)} + \gamma \mathbb{E}_{s' \sim \bar{p}(\cdot|s,a)}[U(s')]] > \mathcal{T}_{\text{SR-KL}}U(s) - \epsilon \tag{19}$$

Thus, we have,

$$0 \leq \mathcal{T}_{\text{SR-KL}}U(s) - \mathcal{T}_{\text{SR-KL}}V(s)$$

$$< (\mathbb{E}_{a \sim \hat{\pi}(\cdot|s)}[r(s,a) - \tau \log \frac{\hat{\pi}(\cdot|s)}{\bar{\pi}(\cdot|s)} + \gamma \mathbb{E}_{s' \sim \bar{p}(\cdot|s,a)}[U(s')]] + \epsilon)$$

$$- (\sup_{\pi \in \Pi} \mathbb{E}_{a \sim \pi(\cdot|s)}[r(s,a) - \tau \log \frac{\pi(\cdot|s)}{\bar{\pi}(\cdot|s)} + \gamma \mathbb{E}_{s' \sim \bar{p}(\cdot|s,a)}[V(s')]])$$

$$\leq (\mathbb{E}_{a \sim \hat{\pi}(\cdot|s)}[r(s,a) - \tau \log \frac{\hat{\pi}(\cdot|s)}{\bar{\pi}(\cdot|s)} + \gamma \mathbb{E}_{s' \sim \bar{p}(\cdot|s,a)}[U(s')]] + \epsilon) \tag{20}$$

$$- (\mathbb{E}_{a \sim \hat{\pi}(\cdot|s)}[r(s,a) - \tau \log \frac{\hat{\pi}(\cdot|s)}{\bar{\pi}(\cdot|s)} + \gamma \mathbb{E}_{s' \sim \bar{p}(\cdot|s,a)}[V(s')]])$$

$$= \mathbb{E}_{a \sim \hat{\pi}(\cdot|s), s' \sim \bar{p}(\cdot|s,a)}[\gamma U(s')] - \mathbb{E}_{a \sim \hat{\pi}(\cdot|s), s' \sim \bar{p}(\cdot|s,a)}[\gamma V(s')] + \epsilon$$

$$\leq \gamma \|U - V\| + \epsilon$$

Applying a similar argument for the case $\mathcal{T}_{\text{SR-KL}}U(s) \leq \mathcal{T}_{\text{SR-KL}}V(s)$ results in

$$|\mathcal{T}_{\text{SR-KL}}U - \mathcal{T}_{\text{SR-KL}}V| < \gamma \|U - V\| + \epsilon. \tag{21}$$

Since $\epsilon$ is an arbitrary positive number, we establish the result, i.e.,

$$|\mathcal{T}_{\text{SR-KL}}U - \mathcal{T}_{\text{SR-KL}}V| \leq \gamma \|U - V\|. \tag{22}$$

$\square$

## G  ENTROPY-REGULARIZED POLICY EVALUATION

This section describes: (1) modification to the TD update for the expected return to optimize for the entropy-regularized expected return, (2) additional modification to account for robustness.

We start with (1).

The entropy-regularized value function is defined as:

$$V_{\text{KL}}^{\pi}(s; \bar{\pi}) \quad = \quad \mathbb{E}^{\pi}[\sum_{t=0}^{\infty} \gamma^t (r_t - \tau \text{KL}(\pi(\cdot|s_t) \| \bar{\pi}(\cdot|s_t)))|s_0 = s] \tag{23}$$

and the corresponding entropy-regularized action value function is given by:

$$Q_{\text{KL}}^{\pi}(s, a; \bar{\pi}) \quad = \quad \mathbb{E}^{\pi}[\sum_{t=0}^{\infty} \gamma^t (r_t - \tau \text{KL}(\pi(\cdot|s_t) \| \bar{\pi}(\cdot|s_t)))|s_0 = s, a_0 = a] \tag{24}$$

$$= \quad r(s, a) - \tau \text{KL}(\pi(\cdot|s) \| \bar{\pi}(\cdot|s)) + \mathbb{E}_{s' \sim p(\cdot|s,a)}[V_{\text{KL}}^{\pi}(s'; \bar{\pi})] \tag{25}$$

Next, we define:

$$\widetilde{Q}_{\text{KL}}^{\pi}(s, a; \bar{\pi}) = r(s, a) + \mathbb{E}_{s' \sim p(\cdot|s,a)}[V_{\text{KL}}^{\pi}(s'; \bar{\pi})] \tag{26}$$

thus,

$$Q_{\text{KL}}^{\pi}(s, a; \bar{\pi}) = \widetilde{Q}_{\text{KL}}^{\pi}(s, a; \bar{\pi}) - \tau \text{KL}(\pi(\cdot|s) \| \bar{\pi}(\cdot|s))) \tag{27}$$

Therefore, we have the following relationship:

$$V_{\text{KL}}^{\pi}(s'; \bar{\pi}) = \mathbb{E}_{a \sim \pi(\cdot|s)} \left[ Q_{\text{KL}}^{\pi}(s, a; \bar{\pi}) \right] = \mathbb{E}_{a \sim \pi(\cdot|s)} \left[ \widetilde{Q}_{\text{KL}}^{\pi}(s, a; \bar{\pi}) - \tau \text{KL}(\pi(\cdot|s) \| \bar{\pi}(\cdot|s)) \right] \tag{28}$$

We now retrieve the TD update for the entropy-regularized action value function:

$$\begin{aligned} \delta_t &= r_t - \tau \text{KL}(\pi(\cdot|s_t) \| \bar{\pi}(\cdot|s_t)) + \gamma Q_{\text{KL}}^{\pi}(s_{t+1} \sim P(\cdot|s_t, a_t), a_{t+1} \sim \pi(\cdot|s_{t+1}); \bar{\pi}) \\ &\quad - Q_{\text{KL}}^{\pi}(s_t, a_t; \bar{\pi}) \\ &= r_t - \tau \text{KL}(\pi(\cdot|s_t) \| \bar{\pi}(\cdot|s_t)) + \gamma Q_{\text{KL}}^{\pi}(s_{t+1} \sim P(\cdot|s_t, a_t), a_{t+1} \sim \pi(\cdot|s_{t+1}); \bar{\pi}) \\ &\quad - \widetilde{Q}_{\text{KL}}^{\pi}(s_t, a_t; \bar{\pi}) + \tau \text{KL}(\pi(\cdot|s_t) \| \bar{\pi}(\cdot|s_t)) \\ &= r_t + \gamma Q_{\text{KL}}^{\pi}(s_{t+1} \sim P(\cdot|s_t, a_t), a_{t+1} \sim \pi(\cdot|s_{t+1}); \bar{\pi}) - \widetilde{Q}_{\text{KL}}^{\pi}(s_t, a_t; \bar{\pi}) \\ &= r_t + \gamma \left[ \widetilde{Q}_{\text{KL}}^{\pi}(s_{t+1} \sim P(\cdot|s_t, a_t), a_{t+1} \sim \pi(\cdot|s_{t+1}); \bar{\pi}) \right. \\ &\quad \left. - \tau \text{KL}(\pi(\cdot|s_{t+1} \sim P(\cdot|s_t, a_t)) \| \bar{\pi}(\cdot|s_{t+1} \sim P(\cdot|s_t, a_t))) \right] - \widetilde{Q}_{\text{KL}}^{\pi}(s_t, a_t; \bar{\pi}) \end{aligned} \tag{29}$$

Note that in the above TD update we replaced $Q_{\text{KL}}^{\pi}$ with $\widetilde{Q}_{\text{KL}}^{\pi}$.

Next, we move to (2).

Before extending the TD update to the robust case, we first consider the *robust* entropy-regularized value function, which is defined as:

$$V_{\text{R-KL}}^{\pi}(s; \bar{\pi}) \quad = \quad \inf_{p \in \mathcal{P}} \mathbb{E}^{p, \pi}[\sum_{t=0}^{\infty} \gamma^t (r_t - \tau \text{KL}(\pi(\cdot|s_t) \| \bar{\pi}(\cdot|s_t)))|s_0 = s] \tag{30}$$

Applying similar steps as above yields the following TD update:

$$
\begin{aligned}
\delta_t &= r_t - \tau \mathrm{KL}(\pi(\cdot|s_t)\|\bar{\pi}(\cdot|s_t)) + \gamma \inf_{p \in \mathcal{P}} Q^{\pi}_{\text{R-KL}}(s_{t+1} \sim p(\cdot|s_t, a_t), a_{t+1} \sim \pi(\cdot|s_{t+1}); \bar{\pi}) \\
&\quad - Q^{\pi}_{\text{R-KL}}(s_t, a_t; \bar{\pi}) \\
&= r_t - \tau \mathrm{KL}(\pi(\cdot|s_t)\|\bar{\pi}(\cdot|s_t)) + \gamma \inf_{p \in \mathcal{P}} Q^{\pi}_{\text{R-KL}}(s_{t+1} \sim p(\cdot|s_t, a_t), a_{t+1} \sim \pi(\cdot|s_{t+1}); \bar{\pi}) \\
&\quad - \widetilde{Q}^{\pi}_{\text{R-KL}}(s_t, a_t; \bar{\pi}) + \tau \mathrm{KL}(\pi(\cdot|s_t)\|\bar{\pi}(\cdot|s_t)) \\
&= r_t + \gamma \inf_{p \in \mathcal{P}} Q^{\pi}_{\text{R-KL}}(s_{t+1} \sim p(\cdot|s_t, a_t), a_{t+1} \sim \pi(\cdot|s_{t+1}); \bar{\pi}) - \widetilde{Q}^{\pi}_{\text{R-KL}}(s_t, a_t; \bar{\pi}) \\
&= r_t + \gamma \inf_{p \in \mathcal{P}} \left[ \widetilde{Q}^{\pi}_{\text{R-KL}}(s_{t+1} \sim p(\cdot|s_t, a_t), a_{t+1} \sim \pi(\cdot|s_{t+1}); \bar{\pi}) \right. \\
&\quad \left. - \tau \mathrm{KL}(\pi(\cdot|s_{t+1} \sim p(\cdot|s_t, a_t))\|\bar{\pi}(\cdot|s_{t+1} \sim p(\cdot|s_t, a_t))) \right] - \widetilde{Q}^{\pi}_{\text{R-KL}}(s_t, a_t; \bar{\pi})
\end{aligned}
\tag{31}
$$

| Hyperparameters | SVG |
|---|---|
| Policy net | 200-200-200 |
| Q function net | 500-500-500 |
| Discount factor ($\gamma$) | 0.99 |
| Adam learning rate | 0.0003 |
| Replay buffer size | 1000000 |
| Target network update period | 200 |
| Batch size | 1024 |
| Activation function | elu |
| Tanh on output of layer norm | Yes |
| Layer norm on first layer | Yes |
| Tanh on Gaussian mean | Yes |
| Min variance | 0.1 |
| Max variance | unbounded |

Table 1: Hyperparameters for SVG

## H  EXPERIMENTS

### H.1  ADDITIONAL DETAILS ON THE SVG BASELINE

For the stochastic value gradients SVG(0) baseline we use the same policy parameterization as for our algorithm, e.g. we have

$$\pi_\theta = \mathcal{N}(\mu_\theta(s), \sigma_\theta^2(s)I),$$

where $I$ denotes the identity matrix and $\sigma_\theta(s)$ is computed from the network output via a softplus activation function.

To obtain a baseline that is, in spirit, similar to our algorithm we used SVG in combination with Entropy regularization. That is, we optimize the policy via gradiend ascent, following the reparameterized gradient for a given state s sampled from the replay:

$$\nabla_\theta \mathbb{E}_{\pi_\theta(a|s)}[Q(a, s)] + \alpha \mathrm{H}\Big(\pi_\theta(a|s)\Big), \tag{32}$$

which can be computed, using the reparameterization trick, as

$$\mathbb{E}_{\zeta \sim \mathcal{N}(0,I)}[\nabla_\theta g_\theta(s, \zeta) \nabla_g Q(g_\theta(s, \zeta), s)] + \alpha \nabla_\theta \mathrm{H}\Big(\pi_\theta(a|s)\Big), \tag{33}$$

where $g_\theta(s, \zeta) = \mu_\theta(s) + \sigma_\theta() * \zeta$ is now a deterministic function of a sample from the standard multivariate normal distribution. See e.g. Heess et al. (2015a) (for SVG) as well as Rezende et al. (2014); Kingma & Welling (2013) (for the reparameterization trick) for a detailed explanation.

### H.2  EXPERIMENT DETAILS FOR MPO AND SVG

In this section we outline the details on the hyperparameters used for the MPO and SVG algorithms. All experiments use a feed-forward two layer neural network with 50 neurons to map the current state of the network to the mean and diagonal covariance of the Gaussian policy. The policy is given by a Gaussian distribution with a diagonal covariance matrix. The neural network outputs the mean $\mu = \mu(s)$ and diagonal Cholesky factors $A = A(s)$, such that $\Sigma = AA^T$. The diagonal factor $A$ has positive diagonal elements enforced by the softplus transform $A_{ii} \leftarrow \log(1 + \exp(A_{ii}))$ to ensure positive definiteness of the diagonal covariance matrix. Tables 2 and 1 show the hyperparameters used for the MPO and SVG algorithms.

### H.3  UNCERTAINTY SET PARAMETERS

Table 3 contains the chosen uncertainty set values for each of the domains and the corresponding holdout set perturbations. The final column of the table contains the parameter that was perturbed.

| Hyperparameters | MPO |
|---|---|
| Policy net | 200-200-200 |
| Number of actions sampled per state | 15 |
| Q function net | 500-500-500 |
| $\epsilon$ | 0.1 |
| $\epsilon_\mu$ | 0.01 |
| $\epsilon_\Sigma$ | 0.00001 |
| Discount factor ($\gamma$) | 0.99 |
| Adam learning rate | 0.0003 |
| Replay buffer size | 1000000 |
| Target network update period | 200 |
| Batch size | 1024 |
| Activation function | elu |
| Layer norm on first layer | Yes |
| Tanh on output of layer norm | Yes |
| Tanh on Gaussian mean | No |
| Min variance | Zero |
| Max variance | unbounded |

Table 2: Hyperparameters for MPO

Table 3: The parameters chosen for the uncertainty set perturbations as well as the holdout set perturbations. The final column contains the parameter that was perturbed.

| Domain | Uncertainty Set Perturbations | Hold-out Test Perturbations | Parameter |
|---|---|---|---|
| Acrobot | 1.0, 1.025, 1.05 meters | 1.15, 1.2, 1.25 meters | First pole length |
| Cartpole Balance | 0.5, 1.9, 2.1 meters | 2.0, 2.2, 2.3 meters | Pole length |
| Cartpole Swingup | 1.0, 1.4, 1.7 meters | 1.2, 1.5, 1.8 meters | Pole Length |
| Cheetah Run | 0.4, 0.45, 0.5 meters | 0.3, 0.325, 0.35 meters | Torso Length |
| Hopper Hop | -0.32, -0.33, -0.34 meters | -0.4, -0.45, -0.5 meters | Calf Length |
| Hopper Stand | -0.32, -0.33, -0.34 meters | -0.4, -0.475, -0.5 meters | Calf Length |
| Pendulum Swingup | 1.0, 1.1, 1.4 Kg | 1.5, 1.6, 1.7 Kg | Ball Mass |
| Walker Run | 0.225, 0.2375, 0.25 meters | 0.35, 0.375, 0.4 meters | Thigh Lengths |
| Walker Walk | 0.225, 0.2375, 0.25 meters | 0.35, 0.375, 0.4 meters | Thigh Lengths |
| Shadow hand | 0.025, 0.022, 0.02 meters | 0.021, 0.018, 0.015 meters | Half-cube width |
| Cartpole Balance: Larger Test Set | 0.5, 1.9 ,2.1 meters | 0.5, 0.7, 0.9, 1.1, 1.3, 1.5, 1.7, 1.9 meters | Pole Length |
| Pendulum Swingup: Larger Test Set | 1.0, 1.1, 1.4 meters | 1.0, 1.1, 1.2, 1.3, 1.4, 1.5 meters | Pole Length |
| Pendulum Swingup - Offline datasets | 1.0, 1.1, 1.2 Kg | 1.5, 1.6, 1.7 Kg | Ball Mass |
| Cartpole Swingup - Offline datasets | 1.0, 1.4, 1.7 meters | 1.2, 1.5, 1.8 meters | Pole Length |

## H.4 MAIN EXPERIMENTS

This section contains two sets of plots. Figure 6 contains bar plots comparing the performance of RE-MPO (blue bars), SRE-MPO (green bars) and E-MPO (red bars) across nine Mujoco domains. The performance of the agents as a function of evaluation steps is shown in Figure 7 for all nine domains respectively. Figrue 8 shows the bar plots for R-MPO, SR-MPO and MPO and Figure 9 shows the corresponding performance of the agents as a function of evaluation steps.

## H.5 INVESTIGATIVE EXPERIMENTS

This section contains additional investigative experiments that were mentioned in the main paper.

Figure 11 presents the difference in performance between the entropy-regularized agents and the **non** entropy-regularized agents agents. Although the performance is comparable (left figure), the entropy-regularized version performs no worse on average than the non-entropy-regularized agent. In addition, there are some tasks where there is a large improvement in performance, such as the

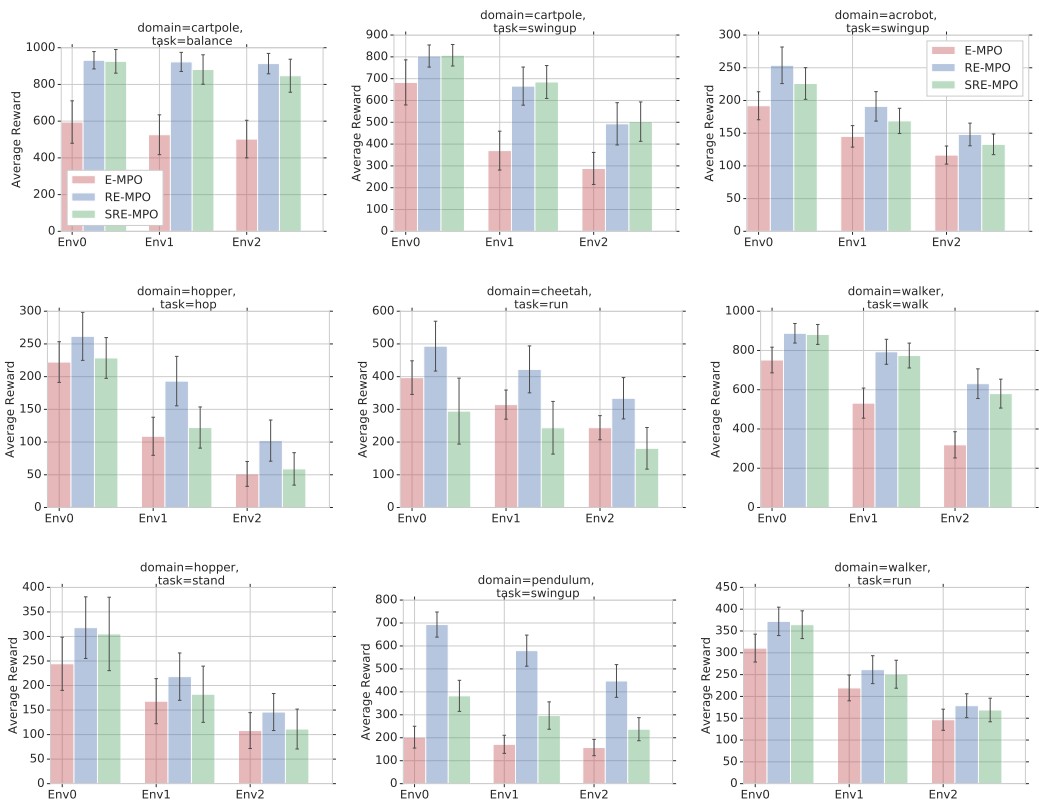

Figure 6: All nine domains showing RE-MPO (blue), SRE-MPO (green) and E-MPO (red).

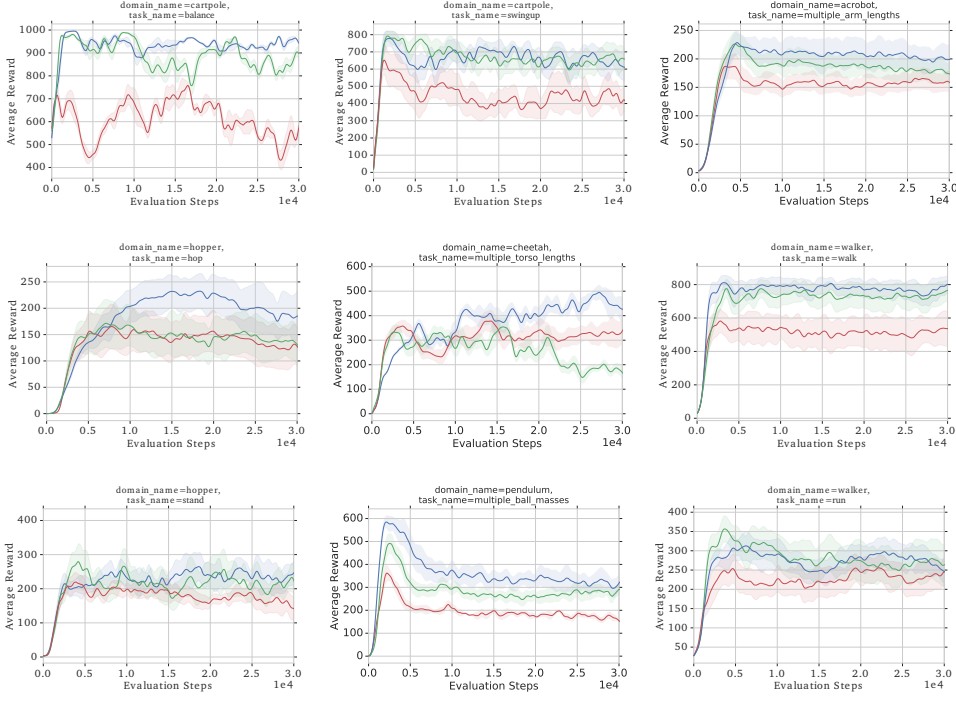

Figure 7: All nine domains showing RE-MPO (blue), SRE-MPO (green) and E-MPO (red) as a function of evaluation steps during training.

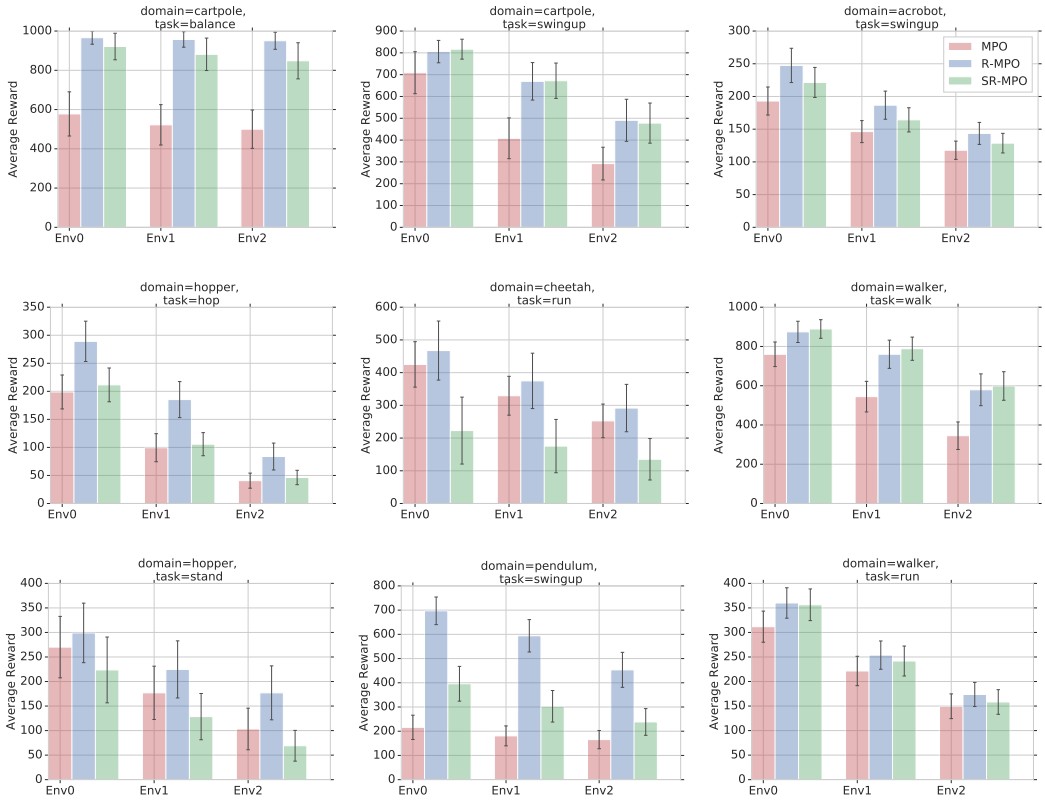

Figure 8: All nine domains showing R-MPO (blue), SR-MPO (green) and MPO (red).

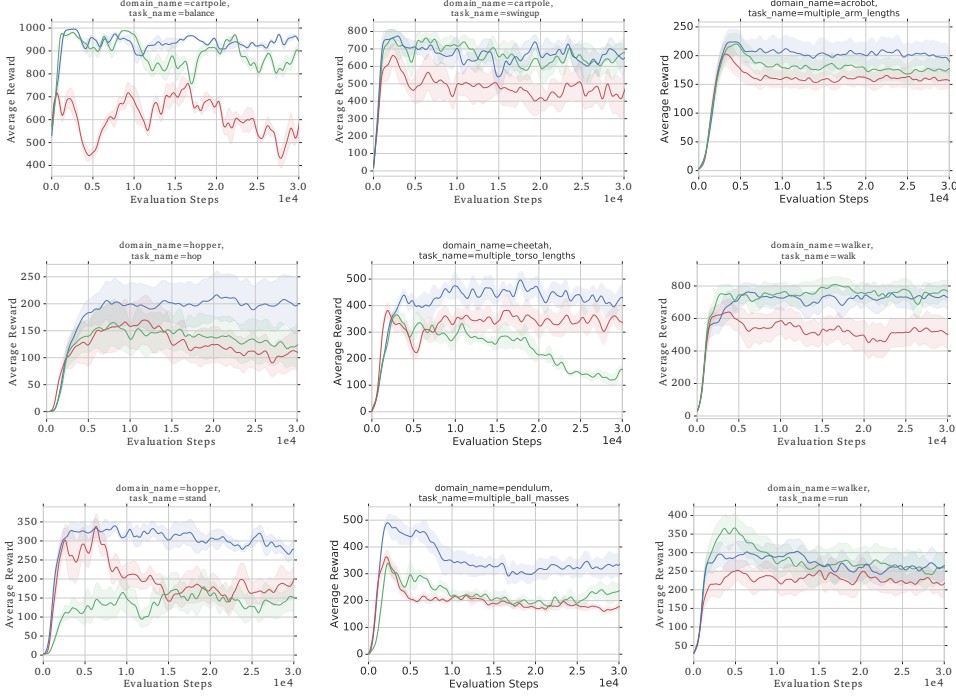

Figure 9: All nine domains showing R-MPO (blue), SR-MPO (green) and MPO (red) as a function of evaluation steps during training.

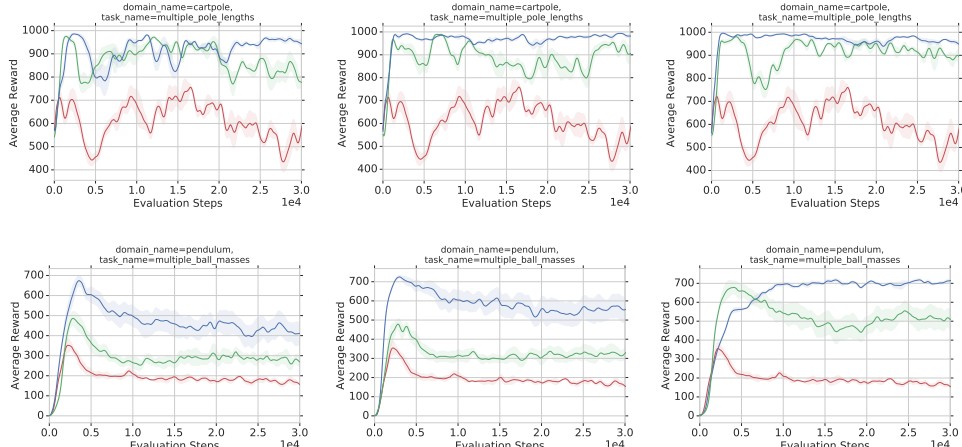

Figure 10: Increasing the range of the training uncertainty set for Cartpole balance (top row) and Pendulum swingup (bottom row).

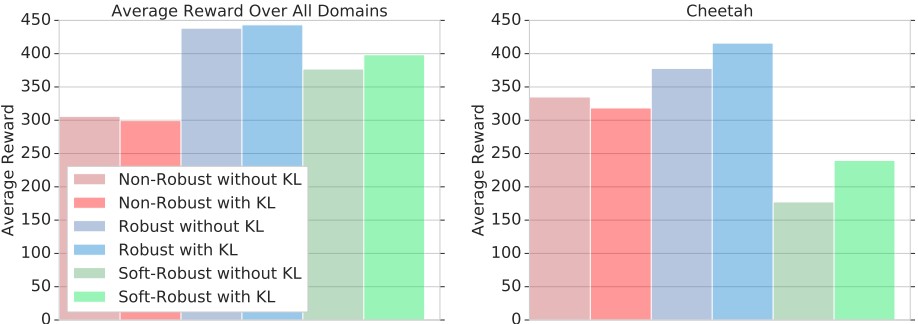

Figure 11: Comparing entropy-regularized objective to the non-entropy regularized objective (left figure). The entropy-regularized version does no worse than the non entropy-regularized setup and in some cases, for example Cheetah, performs considerably better than the expected return objective (right figure).

Cheetah task for the entropy-regularized agent variants non entropy-regularized agent variants (right figure).

**Training with more samples**: Adding three times more samples to the non-robust baseline still yields significantly inferior performance compared to that of the robust and soft-robust versions as seen in Figure 12 for Cartpole balance and Pendulum swingup respectively.

**What about Domain Randomization?** The DR results are shown in Figure 13. As can be seen in the figure, RE-MPO makes better use of a limited number of perturbations compared to *Limited*-DR in Cartpole Balance (left) and Pendulum Swingup (middle) respectively. If the number of perturbations are increased to 100 (right figure) for Pendulum Swingup, DR, which uses approximately 30 times more perturbations, improves but still does not outperform RE-MPO.

**Modifying the uncertainty set**: Figure 14 contains the performance for cartpole balance (top row) and pendulum swingup (bottom row) when modifying the uncertainty set. For the Cartpole Balance task, the original uncertainty set training values are $0.5, 1.4$ and $2.1$ meters for the cartpole arm length. We modified the third perturbation ($2.1$ meters) of the uncertainty set to pole lengths of $1.5, 2.5$ and $3.5$ meters respectively. The agent is evaluated on pole lengths of $2.0, 2.2$ and $2.3$ meters respectively. As seen in the top row of Figure 14, as the training perturbation is near the evaluation set, the performance of the robust and soft-robust agents are near optimal. However, as the perturbation increases further (i.e., $3.5$ meters), there is a drop in robustness performance. This

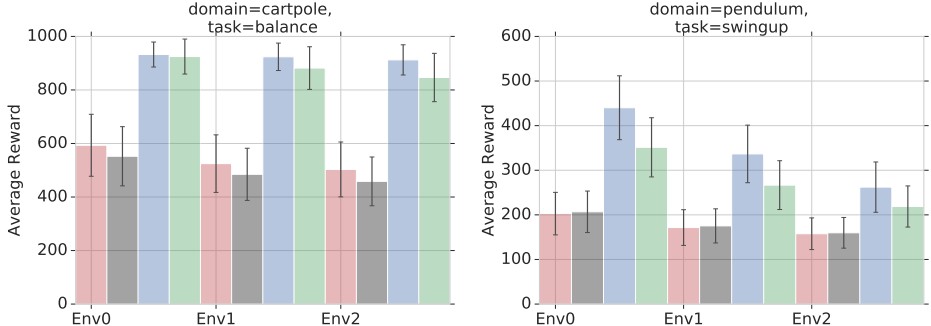

Figure 12: Additional Training Samples: Two plots show 3 times more additional training samples for non-robust E-MPO (dark grey) in the Cartpole Balance and Pendulum Swingup tasks respectively.

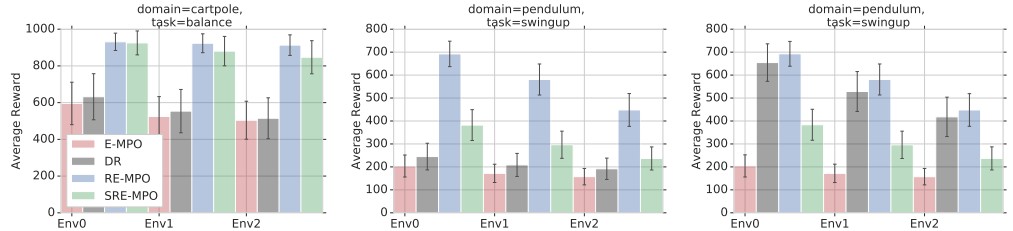

Figure 13: Domain Randomization (DR): Domain randomization performance for the Cartpole balance (left) and Pendulum swingup (middle) tasks. As we increase the number of perturbations for DR to 100 (right figure), we see that performance improves but still does not outperform RE-MPO, which still only uses 3 perturbations.

is probably due to the agent learning a policy that is robust with respect to perturbations that are relatively far from the unseen evaluation set. However, the agent still performs significantly better than the non-robust baseline in each case. For Pendulum Swingup, the original uncertainty set values of the pendulum arm are $1.0, 1.1$ and $1.4$ meters. We modified the final perturbation to values of $1.2, 1.3$ and $2.0$ meters respectively. The agent is evaluated on unseen lengths of $1.5, 1.6$ and $1.7$ meters. A significant increase in performance can be seen in the bottom row of Figure 14 as the third perturbation approaches that of the unseen evaluation environments. Thus it appears that if the agent is able to approximately capture the dynamics of the unseen test environments within the training set, then the robust agent is able to adapt to the unseen test environments. Figure 10 presents the evaluation curves for the corresponding Cartpole Balance (top row) and Pendulum swingup (bottom row) tasks as the third perturbation of the uncertainty set is modified.

**Different Nominal Models**: Figure 15 indicates the effect of changing the nominal model to the median and largest perturbation from the uncertainty set for the Cartpole balance (top row) and Pendulum swingup (bottom row) tasks respectively. For Cartpole, since the median and largest perturbations are significantly closer to the evaluation set, performance of the non-robust, robust and soft-robust agents are comparable. However, for Pendulum swingup, the middle actor is still far from the evaluation set and here the robust agent significantly outperforms the non-robust agent.

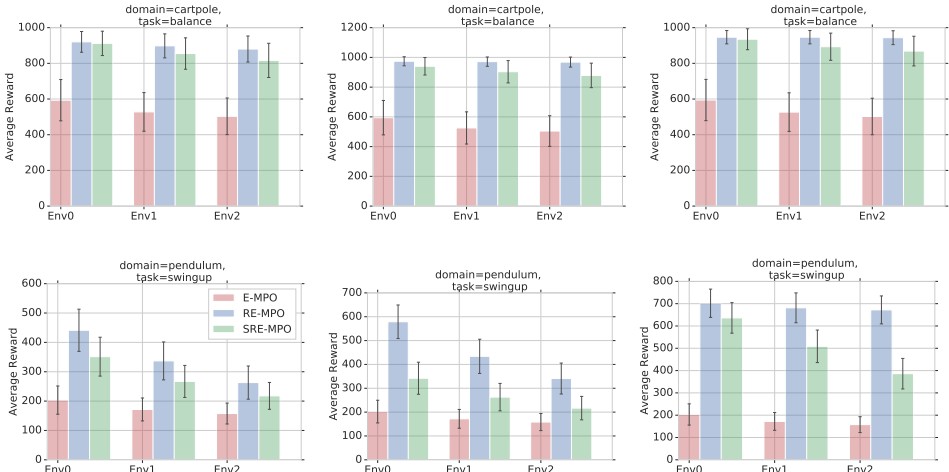

Figure 14: Modifying the uncertainty set: The top row indicates the change in performance for Cartpole balance as the third perturbation of the uncertainty set is modified to $1.5$, $2.5$ and $3.5$ meters respectively. The bottom row shows the performance for Pendulum Swingup for final perturbation changes of $1.2$, $1.3$ and $2.0$ meters respectively.

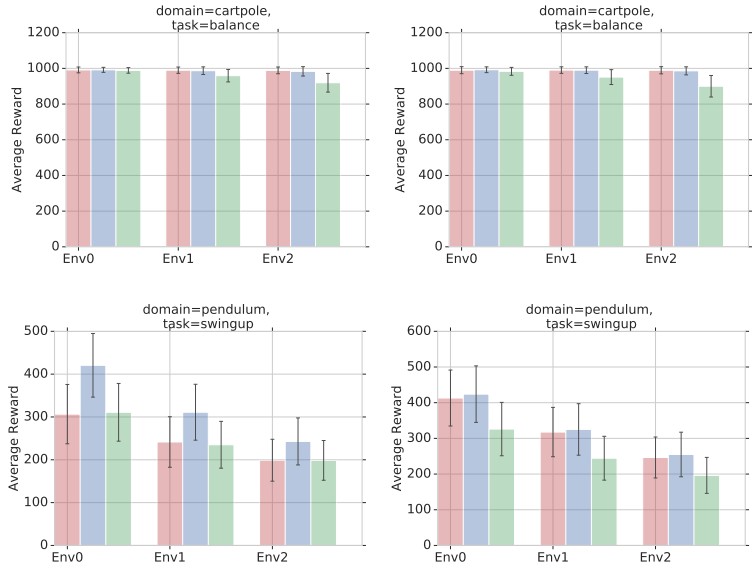

Figure 15: Changing the nominal model: The top two figures indicate setting the nominal model as the median and largest perturbation of the uncertainty set for Cartpole Balance respectively. The right two figures are the same setting but for the Pendulum swingup domain. Legend: E-MPO (red), RE-MPO (blue), SRE-MPO (green).

---

**Algorithm 1** Robust MPO (R-MPO) algorithm for a single iteration

---

1: **given** batch-size (K), number of actions (N), old-policy $\pi_k$ and replay-buffer
2: **// Step 1: Perform policy evaluation on $\pi_k$ to yield $Q_\theta^{\pi_k}$**
3:

$$\min_\theta \left( r_t + \gamma \inf_{p \in \mathcal{P}(s_t, a_t)} \left[ Q_{\hat{\theta}}^{\pi_k}(s_{t+1} \sim p(\cdot|s_t, a_t), a_{t+1} \sim \pi_k(\cdot|s_{t+1})) \right] - Q_\theta^{\pi_k}(s_t, a_t) \right)^2 \,,$$

4: **repeat**
5:     Sample batch of size N from replay buffer
6:     **// Step 2: sample based policy (weights)**
7:     $q(a_i|s_j) = q_{ij}$, **computed as:**
8:     **for** j = 1,...,$K$ **do**
9:       **for** i = 1,...,$N$ **do**
10:         $a_i \sim \pi_k(a|s_j)$
11:         $Q_{ij} = Q^{\pi_k}(s_j, a_i)$
12:         $q_{ij} = $ **Compute Weights**($\{Q_{ij}\}_{i=1...N}$) {See (Abdolmaleki et al., 2018b)}
13:     **// Step 3: update parametric policy**
14:     Given the data-set $\{s_j, (a_i, q_{ij})_{i=1...N}\}_{j=1...K}$
15:     **Update the Policy by finding**
16:     $\pi_{k+1} = \text{argmax}_\pi \sum_j^K \sum_i^N q_{ij} \log \pi(a_i|s_j)$
17:     **(subject to additional (KL) regularization)**
18: **until** Fixed number of steps
19: return $\pi_{k+1}$

---

# I ALGORITHM

The Robust MPO algorithm is defined as Algorithm 1. The algorithm can be divided into three steps: Step (1) perform policy evaluation on the policy $\pi_k$; Step (2) build a proposal distribution $q(a|s)$ from the action value function $Q_\theta^{\pi_k}$; Step (3) update the policy by minimizing the KL divergence between the proposal distribution $q$ and the policy $\pi$. The corresponding robust entropy-regularized version can be seen in Algorithm 2 and the soft-robust entropy-regularized version in Algorithm 3.

---

**Algorithm 2** Robust Entropy-Regularized MPO (RE-MPO) algorithm for a single iteration

---

1: **given** batch-size (K), number of actions (N), old-policy $\pi_k$ and replay-buffer
2: **// Step 1: Perform policy evaluation on $\pi_k$ to yield $Q_\theta^{\pi_k}$**
3:

$$
\min_\theta \Bigg( r_t + \gamma \inf_{p \in \mathcal{P}(s_t, a_t)} \Bigg[ \widetilde{Q}_{\text{R-KL},\hat{\theta}}^{\pi_k}(s_{t+1} \sim p(\cdot|s_t, a_t), a_{t+1} \sim \pi_k(\cdot|s_{t+1}); \bar{\pi})
$$

$$
- \tau \text{KL}(\pi_k(\cdot|s_{t+1} \sim p(\cdot|s_t, a_t)) \| \bar{\pi}(\cdot|s_{t+1} \sim p(\cdot|s_t, a_t))) \Bigg] - \widetilde{Q}_{\text{KL},\theta}^{\pi_k}(s_t, a_t; \bar{\pi}) \Bigg)^2,
$$

4: **repeat**
5:     Sample batch of size N from replay buffer
6:     **// Step 2: sample based policy (weights)**
7:     $q(a_i|s_j) = q_{ij}$, **computed as:**
8:     **for** j = 1,...,$K$ **do**
9:       **for** i = 1,...,$N$ **do**
10:         $a_i \sim \pi_k(a|s_j)$
11:         $Q_{ij} = Q^{\pi_k}(s_j, a_i)$
12:         $q_{ij} = $ **Compute Weights**($\{Q_{ij}\}_{i=1...N}$) {see (Abdolmaleki et al., 2018b)}
13:     **// Step 3: update parametric policy**
14:     Given the data-set $\{s_j, (a_i, q_{ij})_{i=1...N}\}_{j=1...K}$
15:     **Update the Policy by finding**
16:     $\pi_{k+1} = \text{argmax}_\pi \sum_j^K \sum_i^N q_{ij} \log \pi(a_i|s_j)$
17:     **(subject to additional (KL) regularization)**
18: **until** Fixed number of steps
19: **return** $\pi_{k+1}$

---

---

**Algorithm 3** Soft-Robust Entropy-Regularized MPO (SRE-MPO) algorithm for a single iteration

---

1: **given** batch-size (K), number of actions (N), old-policy $\pi_k$ and replay-buffer
2: **// Step 1: Perform policy evaluation on $\pi_k$ to yield $Q_\theta^{\pi_k}$**
3:

$$
\min_\theta \Bigg( r_t + \gamma \Bigg[ \widetilde{Q}_{\text{R-KL},\hat{\theta}}^{\pi_k}(s_{t+1} \sim \bar{p}(\cdot|s_t, a_t), a_{t+1} \sim \pi_k(\cdot|s_{t+1}); \bar{\pi})
$$

$$
- \tau \text{KL}(\pi_k(\cdot|s_{t+1} \sim \bar{p}(\cdot|s_t, a_t)) \| \bar{\pi}(\cdot|s_{t+1} \sim \bar{p}(\cdot|s_t, a_t))) \Bigg] - \widetilde{Q}_{\text{KL},\theta}^{\pi_k}(s_t, a_t; \bar{\pi}) \Bigg)^2,
$$

4: **repeat**
5:     Sample batch of size N from replay buffer
6:     **// Step 2: sample based policy (weights)**
7:     $q(a_i|s_j) = q_{ij}$, **computed as:**
8:     **for** j = 1,...,$K$ **do**
9:       **for** i = 1,...,$N$ **do**
10:         $a_i \sim \pi_k(a|s_j)$
11:         $Q_{ij} = Q^{\pi_k}(s_j, a_i)$
12:         $q_{ij} = $ **Compute Weights**($\{Q_{ij}\}_{i=1...N}$) {see (Abdolmaleki et al., 2018b)}
13:     **// Step 3: update parametric policy**
14:     Given the data-set $\{s_j, (a_i, q_{ij})_{i=1...N}\}_{j=1...K}$
15:     **Update the Policy by finding**
16:     $\pi_{k+1} = \text{argmax}_\pi \sum_j^K \sum_i^N q_{ij} \log \pi(a_i|s_j)$
17:     **(subject to additional (KL) regularization)**
18: **until** Fixed number of steps
19: **return** $\pi_{k+1}$

---

