# OpenReview forum: "Robust Reinforcement Learning for Continuous Control with Model Misspecification"
_ICLR.cc/2020/Conference — Accept (Poster)_

### Official Review · AnonReviewer3 · 2019-10-22
**Official Blind Review #3**

**Rating:** 8

**Review:**

AFTER REBUTTAL:
The authors answered all my questions and (significantly) modified the paper according to all the comments from all reviewers. The paper has significant value given the extensive experimental results and I believe could provide insight to other researchers in the field. I still have (like other reviewers) some concerns about scalability regarding the space of possible perturbations, but, as it is, the paper is worth for publication. For these reasons, *I increase my score to 7*, since I believe it's better than a weak accept (ignore the system rating, since it doesn't allow to put 7).

======================================================
Summary:

The authors provide a framework to incorporate robustness against perturbed dynamics to RL agents. They build upon the MPO algorithm and incorporate a few modifications that enable such robustness and entropy regularization. They show in multiple experiments that the robust versions of MPO outperform the non-robust ones when testing on environments with novel perturbations.


Decision:

I am happy with the experiments conducted on this paper and I think they would be a nice contribution to the community. However, I have serious concerns about the novelty of the proposed approach. The authors claim many novel contributions, but most of them are special cases of existing literature (not cited), or are simple modifications to existing approaches. For this reason my score is a reject. If the authors clearly cite the existing literature, state their contributions accordingly, and redirect the paper more towards their good experimental results (which they have plenty) I would be willing to substantially upgrade my score. Find below my supporting arguments.


About novelty
=============


The authors claim to have a novel framework and novel MDP formulations  for robust MDPs disregarding a substantial volume of the literature on robustness in the context of planning, control, reinforcement learning and MDPs. Since the authors really consider two types of robustness (robustness to dynamics and robustness to action stochasticity) I will expose in the following the relevant literature that is missing on both types, and when combining both.



Robustness to Dynamics:

Bart van den Broek, Wim Wiegerinck, and Hilbert J. Kappen.  Risk sensitive path integral control. In UAI, 2010.

Arnab Nilim and Laurent El Ghaoui.   Robust control of markov decision processes with uncertain transition matrices.Operations Research, 53(5):780–798, 2005

Wolfram Wiesemann, Daniel Kuhn, and Berc  Rustem.  Robust markov decision processes. Mathematics of Operations Research, 38(1):153–183, 2013.

Lars Peter Hansen and Thomas J Sargent.Robustness. Princeton university press, 2008

Yun Shen, Michael J Tobia, Tobias Sommer, and Klaus Obermayer. Risk-sensitive reinforcement learning. Neural computation, 26(7):1298–1328, 2014.

Yinlam  Chow,  Aviv  Tamar,  Shie  Mannor,  and  Marco  Pavone.   Risk-sensitive  and  robust decision-making: a cvar optimization approach.   In Advances in Neural Information Pro-cessing Systems, pages 1522–1530, 2015.



Robustness to Action Stochasticity:

Roy Fox, Ari Pakman, and Naftali Tishby.  G-learning: Taming the noise in reinforcement learning via soft updates.arXiv preprint arXiv:1512.08562, 2015.

Jonathan Rubin, Ohad Shamir, and Naftali Tishby.  Trading value and information in mdps.In Decision Making with Imperfect Decision Makers, pages 57–74. Springer, 2012.

Daniel A Braun, Pedro A Ortega, Evangelos Theodorou, and Stefan Schaal.  Path integral control  and bounded  rationality.   In Adaptive  Dynamic  Programming  And  ReinforcementLearning (ADPRL), 2011 IEEE Symposium on, pages 202–209. IEEE, 2011.



Combination of both:

[1] Grau-Moya, Jordi, et al. "Planning with information-processing constraints and model uncertainty in Markov decision processes." Joint European Conference on Machine Learning and Knowledge Discovery in Databases. Springer, Cham, 2016.


Importantly, in [1] above, it is shown a very similar line of research as the authors are proposing. In [1] it is combined entropy regularization with robustness in dynamics. In particular, using the formulation in [1] one can see how the formulations in the present paper can be recovered by setting the parameters in various ways. For example,
1- letting the entropy regularization coefficient ($\alpha$ in [1]) to a finite value and the robustness coefficient ($\beta$ in [1]) to $-\infty$ one recovers the traditional Robust MDP formulation.
2- Selecting a uniform prior over discrete theta (where each component in theta would correspond to a particular perturbation)  and setting $\beta \rightarrow 0$ one can also recover Soft-RE formulation (although a non-MPO version of it, i.e. just the MDP formulation).
3- Similarly, one can get rid of the entropy regularization by setting $\alpha \rightarrow \infty$.

Note that in [1] the uncertainty set is determined by a KL constraint whereas here it is determined by the chosen perturbations. However, in [1] setting the proper prior and $\beta$, one can obtain the same behaviour.


Importantly, in all previous references multiple contraction proofs (for entropy regularization Bellman operators, for robust operators alone, and for robust and entropy regularized operators) have already been discovered.

I hope all of the previous points can convince the authors about my decision above regarding novelty. Note, please, that I am aware that a robust version of MPO is slightly novel, however, a robust MDP formulation is more general than that and it could have perfectly been applied to any other RL optimization algorithm. Therefore, I don't think adding robustness to MPO alone is sufficient for enough novelty (other wise we would publish 1 paper per method converting one by one all known algorithms into their robust version of it).


About scalability
=================

I liked the experimental section. However, I have a question about practicality of the approach. As the authors show in the paper, the proposed method scales well in terms of  high-dimensional state and action spaces. But what about the scalability in terms of possible perturbations? I can imagine a plausible situation where we are dealing with a high-dimensional robot, let's say a humanoid robot, which allows for many possible "perturbations" e.g. longer left leg, longer right leg, longer fingers, shorter thumbs, and so on. The space of possible perturbations can be potentially very big. Could the authors elaborate on scalability issues of this type? Have they already thought about how one would solve this? I imagine that perturbing, let's say in one dimension, might change the dynamics quite in a different way compared to perturbing in another dimension. It would be interesting to see how their method generalizes when testing on unseen "dimension" perturbations.

Although this might present problems in terms of generalization, I understand that it does not invalidate the results presented here.


Good experimental results:

I have carefully checked all experimental results and I must say that they are very well executed e.g. multiple ablation studies and sufficient environments to support their approach.  Maybe a small comment: In figures 6 and 8 in the appendix there are multiple plots that only differ in the data (the titles are the same, e.g. two hopper figures with the same title). Maybe the authors can correct for this.


**Experience Assessment:**

I have published in this field for several years.

**Review Assessment: Checking Correctness Of Derivations And Theory:**

I carefully checked the derivations and theory.

**Review Assessment: Checking Correctness Of Experiments:**

I carefully checked the experiments.

**Review Assessment: Thoroughness In Paper Reading:**

I read the paper thoroughly.

---

> ### Author Response · Authors · 2019-11-08
> **Response to reviewer 3**
>
> Thank you for your review. We will try and address all of your questions as best we can.
>
> Robustness to Dynamics:
> >> Thank you for the citations. We have included these in the related works section in the revised paper.
>
> “the authors really consider two types of robustness (robustness to dynamics and robustness to action stochasticity)”
> “Robustness to Action Stochasticity:”
> >> While we do not claim to be robust to action stochasticity, only dynamics, we have included the works you suggested to the related works section in the paper.
>
> "Importantly, in [1] above, it is shown a very similar line of research as the authors are proposing."
> "Note that in [1] the uncertainty set is determined by a KL constraint whereas here it is determined by the chosen perturbations. However, in [1] setting the proper prior and , one can obtain the same behaviour."
>
> >> Thank you for your detailed response and the reference. We have cited this work in our paper and have added a discussion about it. It is important to note a number of key differences between this work and our paper:
> 1) The authors of [1] state that their work is a special case of robust MDPs: ‘Our approach falls within the robustness methods that use a restricted set of permissible models because we extremize the biased belief  under the constraint that it has to be within some information bounds measured by the Kullback-Leibler divergence from a reference Bayesian posterior.’. The original Robust MDP setup (which includes our case) does not have such limitations and therefore should be robust to a larger range of perturbations (and therefore permissible models).
> 2) As you mentioned the uncertainty set is determined by a KL constraint. This has the effect of regularizing the reward with respect to model uncertainty. This means that the TD error will regularize the *immediate* reward to take into account model uncertainty rather than *explicitly* select the long term value that yields the worst case return. Therefore the policy in [1] will almost certainly be higher variance and less stable for higher dimensional problems.
> 3) The uncertainty set elements from [1] output a probability distribution over model parameter space whereas the uncertainty set elements in our formulation output a distribution over next states.
> 4) [1] is highly dependent on the reference bayesian posterior distribution.
>
> These works are however very related and we will certainly emphasize that this work is similar to the soft-robust setup.
>
>
> "I am aware that a robust version of MPO is slightly novel, however, a robust MDP formulation is more general than that and it could have perfectly been applied to any other RL optimization algorithm ... sufficient for enough novelty"
> >> Again, we thank you for making us aware of interesting related work. We do think that given our previous response, there is sufficient differences between our setup and [1]. In addition, we needed to reformulate the optimization problem for MPO to ensure that entropy-regularization and robust updates made sense. This is the first application of robustness and entropy-regularization in a continuous control RL algorithm (which includes soft-robustness) and as such, we feel has sufficient novelty, especially when we couple this with the numerous experiments, including that of a high dimensional continuous control robotics hand.
>
> "In figures 6 and 8 in the appendix there are multiple plots that only differ in the data (the titles are the same, e.g. two hopper figures with the same title). Maybe the authors can correct for this."
> >> We have modified this. Thanks for pointing this out.
>
> "Could the authors elaborate on scalability issues of this type?"
> >> Great question. We have thought about this in-depth and actually think that increasing the number of dimensions will actually *help* in our setup to yield a more robust agent. If we perturb a single parameter in one dimension, we have a fairly small space of dynamics perturbations that we are enabling the agent to explore. If we add additional dimensions (e.g., mass, toe length etc), then we are effectively exploring over a volume of possible perturbations in parameter space. This we believe will help with generalization to a larger set of robust policies. We are actively exploring this direction as the next steps.
>
> We appreciate your positive comments regarding the experiments and hope that the above explanations have convinced you of the overall novelty of this paper and its relevance for publication.

---

> > ### Comment · AnonReviewer3 · 2019-11-12
> > **Clarify immediate reward**
> >
> > Thank you for your detailed response and for modifying the paper accordingly. I certainly agree with most of your responses. There is only one argument that I don't fully grasp and it is about the immediate reward. Could you clarify what do you mean by that? In my view [1] captures all future pessimism through their robust value function.  This is so since the KL is added to the Bellman operator, which is used to specify the optimal robust  value function.

---

> > > ### Author Response · Authors · 2019-11-14
> > > **Response to Reviewer 3 (Part 2)**
> > >
> > > “[1] captures all future pessimism through their robust value function”
> > > >> Thank you for your reply. We would firstly like to emphasize that we certainly see [1] as another formulation for learning a robust value function which is capable of capturing future pessimism. While there are subtle, yet important differences between both techniques, the learned policies will yield robust behaviour, which may be slightly different in each case. For example, it would be interesting to test [1] on the Walker Walk task to see if it learns a similar policy to that of R-MPO where the agent ‘drags’ its leg; or whether it learns something fundamentally different, yet still robust.
> > >
> > > *Entropy regularization content:*
> > > As we mentioned to reviewer 2, based on the feedback, we have decided to move the majority of the entropy-regularization discussion to the Appendix and focus more on the empirical results. We refer to entropy-regularization briefly in the Section discussing Robust MPO and only include the main theoretical results. Please let us know if there is anything else you would like us to change. We have also added an additional experiment which we discuss below.
> > >
> > > *Additional experiment:*
> > > Since the emphasis has now been placed more on the empirical analysis of the robustness formulation, we have added an additional experiment to the paper. We have trained the uncertainty set of transition models from offline data, inject them into the nominal simulator, and show that R-MPO with these uncertainty sets is still able to yield robust performance which is significantly better than the non-robust MPO baseline. This mimicks likely real-world scenarios such as in data cooling centers and factory robotics where a nominal simulator may be available as well as datasets for individual cooling units in the datacenter or individual robots in the factory. We analyze the performance of these uncertainty sets trained on varying amounts of data and show that there is a `sweet spot’ whereby this version of R-MPO, which we call Data Driven R-MPO (DDR-MPO)) produces  improved performance over our original R-MPO in two domains. We also provide intuition for this result.

---

### Official Review · AnonReviewer2 · 2019-10-23
**Official Blind Review #2**

**Rating:** 6

**Review:**


# Summary
The paper compares 3 ways to account for the variability of the dynamics model in the TD error computation:
(i) be robust (take the lowest target value obtained when evaluating the value-function on the training distribution of models);
(ii) be Bayesian (average over models);
(iii) domain randomization (compute TD error as usual but randomize the dynamics to obtain a more diverse data set).

The comparison is carried out using the MPO algorithm. An entropy term is furthermore added to the MPO objective to yield a new E-MPO algorithm, but it didn't affect the performance much (cf. Figs. 5-6 and 7-8).

The paper is mainly an empirical study.

# Decision
The engineering effort of running many experiments and ablation studies is appreciated. However, there are big questions to the evaluation methodology and the contribution of the paper, that preclude me from recommending it for publication in its current form.

# Concerns
1) First, my concern is the problem setting and evaluation methodology, and in particular Table 3 with varied parameters. The authors chose to have 3 values during training, and then evaluate the optimized policy on further 3 testing environments. There are many questions to this table. I give only a few.
        - For most environments, training and test ranges are disjoint (e.g., train on 1.00–1.05m and test on 1.15–1.25). Why? Would your results hold if they are not disjoint?
        - Fig. 14 shows that when trained on a single model with parameters closest to the evaluation range (right column), there is no difference between the algorithms on the Cartpole and a minor difference on the Pendulum. That seems to imply that it is sufficient to just train on one environment which is most similar to the evaluation environments to perform well in the experiments. Is it true? Do you have a counterexample?
        - Natural question, of course, what do you propose to do when varying multiple parameters? Discretizing over several values will quickly yield exponential explosion.
        - Did you try using distributions over ranges instead of fixed parameters? That would be a bit more solid than hand-picked values.

2) Evaluation of domain randomization. The authors say that "TD errors are then averaged together" when doing domain randomization. That means, when updating the value function, gradients corresponding to various models get intermixed. It is usually better to fix the domain, then do a few gradient updates using the data from that domain, and after that collect data from another domain. Such procedure is similar to keeping a target network in Q-learning, which is absolutely necessary to stabilize the learning process. It would be interesting to see if domain randomization still performs poorly when applied in this way.

3) It would be helpful to sharpen the main message.
        - Right now there is the E-MPO introduced, but then it is not really compared to plain MPO (only plots are shown in the Appendix); it is only said that it doesn't hurt to have the entropy term. It seems orthogonal to the robustification scheme in the TD error, so maybe it would make sense to omit the entropy for clarity of exposition.
        - What is the conclusion? The fact that robust works better than non-robust is obvious even without any experiments. So, the only question one could have is whether it is better to average or to take the worst-case model. I didn't get it from the paper whether there is any conclusion regarding that.
        - In Conclusion, the authors appeal to some theoretical contribution of this paper, such as proof of contraction in Theorem 1. I would consider it a very minor contribution as it is mainly a recapitulation of known results as both the robust Bellman operator and the entropy-regularized one are known to be contractions. Furthermore, this theorem is not really used in the paper, so to me it seems detached.

Small: typo in Sec. 7.: you say "to lengths of 2.0, 2.1 and 2.2", but according to Table 3, these should be "2.0, 2.2 and 2.3"

# AFTER REBUTTAL
The authors incorporated the feedback of the reviewers and *significantly* rewrote the paper. Now it is more focused and puts the work in context by discussing connections to Grau-Moya etc. I still think the contribution is minor and the method will not scale, but from the formal side, the paper fulfills the requirements to be accepted.

**Experience Assessment:**

I have published one or two papers in this area.

**Review Assessment: Checking Correctness Of Derivations And Theory:**

I assessed the sensibility of the derivations and theory.

**Review Assessment: Checking Correctness Of Experiments:**

I carefully checked the experiments.

**Review Assessment: Thoroughness In Paper Reading:**

I read the paper at least twice and used my best judgement in assessing the paper.

---

> ### Author Response · Authors · 2019-11-08
> **Response to Reviewer 2**
>
> Thank you for your review. We will try and address all of your questions as best we can.
>
> “… training and test ranges are disjoint … Would your results hold if they are not disjoint?”
> >> Our results would hold if they are not disjoint. This should actually make the training easier as you are effectively training the agent closer to the test set distribution. We realized in the experiments that having an uncertainty set of small perturbations ‘near’ the nominal model, that are disjoint from the test set, is actually enough for the agent to understand how perturbations affected the system dynamics. This aids the agent in generalizing to more significant dynamics changes.
>
>
> “Fig. 14 shows ... train ... evaluation environment … a counterexample?”
> >> This is indeed true and was the purpose of Figure 14. The problem in general is that it is unclear, especially on a real robot or building cooling system for example, what the exact evaluation environment dynamics are. That is why our technique is potentially useful since we only need small perturbations from the nominal training environment in our uncertainty set to generalize to a wide variety of environment perturbations.
>
> “Discretizing over several values ... exponential explosion.”
> >>This is correct. What we found is that capturing small perturbations around the nominal model is often enough to generalize to a wide variety of environment perturbations. Therefore a small grid search around the nominal environment should suffice.
>
> “ ... omit the entropy for clarity of exposition.”
> >> We wanted to include entropy regularization because it encourages exploration and helps prevent convergence to sub-optimal policies. We found it to be helpful overall, especially in the Cheetah task (Figure 10). As such, we wanted to provide clarity for readers that this is indeed a contraction operator and therefore any policy evaluation algorithm employing this robust/soft-robust operator will converge to a unique fixed point.
>
>
> “ What is the conclusion? robust vs. non-robust obvious ... average vs. worst-case model.”
> >> I think there are multiple conclusions: (1) We think that robust works better than non-robust is not that obvious for the following reasons. (i) Since we are learning a worst-case value function, it is possible that the agent learns to be overly conservative and therefore doesn’t do anything. We confirmed here that for all the Mujoco problems we evaluated on, this is not the case. (ii) This method is scalable. It is a natural concern that this method might not scale, but as we see for the Shadow Hand experiment, this method scales to significantly higher dimensional problems than Mujoco and therefore can be useful in practice. (iii) Doing data augmentation (i.e., the limited form of domain randomization in Figure 3) was significantly less efficient than our robustness approach. (iv) The robustness policy is actually fundamentally different from that of the regular expected return policy. If you look at the walker Walk video in the Abstract URL, you will see that the walker agent learns to *drag* its leg so as to be robust to the new thigh length. This is a very different behaviour to that of the expected return agent. We can make these conclusions more explicit if you like?
>
> In terms of average (soft-robust) vs. worst-case (robust) model, this was something we pointed out in Figure 2. The average model seems to do well over a small range of perturbations compared to the robust approach (as seen in Figure 2, right and second from right). However, as the perturbations increase, we see that the robust approach maintains a significantly higher level of performance compared to the average approach. So, even though you suffer a dip in performance for small perturbations, our worst-case approach significantly outperforms the average model as the perturbations increase.
>
>  “… proof of contraction in Theorem 1… minor contribution…”
> >> The contraction contribution is still a contribution that is worth mentioning since we do combine entropy regularization and robustness into a single Bellman operator (for the soft-robust case too). This also required us to modify the MPO optimization problem and the corresponding policy evaluation update for MPO to ensure that it is indeed optimizing with respect to this new operator. This ensured that E-MPO still converged and worked as expected.
>
> “this theorem is not really used in the paper”
> >> We probably should have been more explicit, but this Theorem is used in the paper. As a result of the contraction property, we are ensuring that Step 1 in algorithms 1,2 and 3 in the paper converge to a unique fixed point. This ensures that MPO and any other actor critic algorithm that employs this change will still work. So you can view step 1 in each algorithm as effectively applying this Bellman operator repeatedly until convergence. We can make this more explicit in the updated version.
>
> “typo in Sec. 7”
> >> Thank you. It has been amended.

---

> > ### Comment · AnonReviewer2 · 2019-11-13
> > **Response to the authors**
> >
> > Thanks for addressing my concerns. Nevertheless, I still have a few.
> >
> > | This is indeed true and was the purpose of Figure 14.
> > The difficulty I am having is the following. As you said, it is best to train on just one model which is closest to the true model. Then the question is: does choosing the worst out of a fixed set (the mini-max formulation) or averaging over models (the entropy formulation) help at all? It seems that these approaches would only work either if the worst model is closest to the true model or the average model is close to the true one.
> >
> > | What we found is that capturing small perturbations around the nominal model is often enough
> > If there are, e.g., 20 parameters, and one picks 3 values for each, there are 3^20 variations of the environment. This exponential growth seems problematic, which is also mentioned by R3.
> >
> > | We wanted to include entropy regularization because it encourages exploration
> > Looking at the plots for MPO and E-MPO on pages 20 and 21, there is hardly any difference. On the other hand, in the paper there is a lot of discussion of E-MPO. That is why I suggested to omit the discussion of the "E-" version, as it does not provide significant improvement and is orthogonal to the robustification of MPO, which is the main focus of the paper (judging from the title).
> >
> > | We can make these conclusions more explicit if you like?
> > Thank you, this is a really nice summary. Including these points in the paper would definitely be useful.
> >
> > | The contraction contribution is still a contribution that is worth mentioning
> > I am sorry, maybe I was a bit unclear in my review. I do think that this theorem is a valuable result (although R3 mentioned that it may have been derived in prior work). My main concern is that it talks about the "E-" version of MPO, and it interferes with the robustness discussion. You could consider first describing everything about robustness and then making a section that talks about entropy regularization; then you can put the theorem there and do some ablation studies that show that adding entropy regularization further improves upon pure robustness.
> >
> > | As a result of the contraction property, we are ensuring that Step 1 in algorithms 1,2 and 3 in the paper converge to a unique fixed point.
> > Algorithm 1 is R-MPO, and its  Step 1 is assured by  (Iyengar, 2005). In Algs. 2-3, I agree, one can refer to the theorem. As you suggested, mentioning it explicitly in the text would be certainly helpful to the reader.

---

> > > ### Author Response · Authors · 2019-11-14
> > > **Response to Reviewer 2 (Part 2)**
> > >
> > > “| This is indeed true and was the purpose of Figure 14.
> > > The difficulty I am having i”
> > > >> The robust and soft-robust techniques should work well if the dynamics captured in the uncertainty set allow the agent to generalize to unseen dynamics that are within the distribution of the holdout set dynamics. Therefore, it is less important that one of the models is close to the true holdout set model, but rather that the dynamics captured by the uncertainty set models allows for sufficient generalization to holdout set dynamics.
> > >
> > > “| As a result of the contraction property, we are ensuring that Step 1 in algorithms 1,2 and 3 in the paper converge to a unique fixed point… helpful to the reader.”
> > > >> Thank you for the feedback. We have updated the paper to make this more explicit.
> > >
> > > “| We can make these conclusions more explicit if you like?”
> > > >> In the revised version, we have tried to make these conclusions more explicit throughout the paper. We have added a discussion on the Walker Walk policy, in the Mujoco Experiments paragraph, which is fundamentally different in the robust setting as the agent learns to ‘drag’ its leg. We emphasize scalability when referring to the Shadowhand experiment. We also emphasize our methods efficiency compared to data augmentation (i.e, Limited DR).
> > >
> > > “| We wanted to include entropy regularization because it encourages exploration”
> > > “| The contraction contribution is still a contribution that is worth mentioning… adding entropy regularization further improves upon pure robustness.”
> > > >> Thank you for the comments. Based on your feedback, we have moved the majority of the entropy-regularization discussion to the Appendix. We refer to it briefly in the Section discussing Robust MPO and only include the main theoretical results. We agree that the paper has a better flow. Please let us know if there is anything else you would like us to change. We have also added an additional experiment to the 'Investigative Studies' section which we discuss at the end of this response.
> > >
> > > “| What we found is that capturing small perturbations around the nominal model is often enough
> > > If there are, e.g., 20 parameters, and one picks 3 values for each”
> > > >> Thank you for the question. Here is the answer we gave R3: “We have thought about this in-depth and actually think that increasing the number of parameter dimensions will actually *help* in our setup to yield a more robust agent. If we perturb a single parameter in one dimension, we have a fairly small space of dynamics perturbations that we are enabling the agent to explore. If we add additional dimensions (e.g., mass, toe length etc), then we are effectively exploring over a volume of possible perturbations in parameter space. This we believe will help with generalization to a larger set of robust policies. We are actively exploring this direction as the next steps.”
> > >
> > > That being said, we do agree that it would be ideal to find efficient ways of sampling this space. The focus of this initial work was to scale the robustness framework to high dimensional, continuous control domains and analyse the performance. However, as we said to R3, we certainly see scaling the parameter perturbations as an exciting research direction going forward.
> > >
> > > *Additional note to reviewers:*
> > > Since the emphasis has now been placed more on the empirical analysis of the robustness formulation, we have added an additional experiment to the ‘Investigative Studies’ section of paper. We have trained the uncertainty set of transition models from offline data, injected them into the nominal simulator, and show that R-MPO with these uncertainty sets is still able to yield robust performance which is significantly better than the non-robust MPO baseline. This mimicks likely real-world scenarios such as in data cooling centers and factory robotics where a nominal simulator may be available as well as datasets for individual cooling units in the datacenter or individual robots in the factory. We analyze the performance of these uncertainty sets trained on varying amounts of data and show that there is a `sweet spot’ whereby this version of R-MPO, which we call Data Driven R-MPO (DDR-MPO)) produces improved performance over our original R-MPO in two domains. We also provide intuition for this result.

---

> > > > ### Comment · AnonReviewer2 · 2019-11-14
> > > > **Thanks for the effort, I will check the paper**
> > > >
> > > > Thanks for putting the effort quickly and addressing my comments. The paper now looks significantly different, I will take a look at it.

---

> > > > > ### Author Response · Authors · 2019-11-15
> > > > > **Thank you**
> > > > >
> > > > > Thank you for taking another look. We appreciate it.

---

### Official Review · AnonReviewer1 · 2019-10-27
**Official Blind Review #1**

**Rating:** 6

**Review:**



Rebutal Response:
I keep my rating of weak accept due to the extensive empirical evaluation. However, I want to point out that the Area chair should assign more weight to the other reviews as the other reviewers have provided more extensive reviews and have more knowledge about the existing literature.


###########
Summary:
The paper proposes a robust variant of MPO and evaluates the performance on control tasks including the cartpole, hopper, walker & the shadow hand. The performance shows that the robust MPO outperforms vanilla MPO and MPO + Domain randomization.

Open Questions:
- I am not convinced by the domain randomization performance as the performance increases only marginally over the vanilla MPO and DR has usually performed quite well for robust policies. Can you explain this marginal increase?

- Could the authors please include a qualitative evaluation of the learned controllers? Especially as the Cartpole stabilization  is a linear system one could compare against the analyitc optimal controllers.

Conclusion:
Currently, I would rate the paper as weak accept, as the derivation is interesting and the approach seems to work quite well in  simulation. However, the work is only incremental and does not propose a new perspective to robust RL. It would be nice to show that the policy is also robust for a physical system.

Minor Comments
- What does the bar mean, e.g., \bar{J} & \bar{\pi} mean? I cannot find a definition of the bar.
- Typo in the Background section 'Kullback-Liebler  (KL)'

(I can try to extend my review if the other reviewers disagree)

**Experience Assessment:**

I have read many papers in this area.

**Review Assessment: Checking Correctness Of Derivations And Theory:**

I assessed the sensibility of the derivations and theory.

**Review Assessment: Checking Correctness Of Experiments:**

I assessed the sensibility of the experiments.

**Review Assessment: Thoroughness In Paper Reading:**

I made a quick assessment of this paper.

---

> ### Author Response · Authors · 2019-11-08
> **Response to Reviewer 1**
>
> Thank you for your review. We will try and address all of your questions as best we can.
>
> "I am not convinced by the domain randomization performance as the performance increases only marginally over the vanilla MPO and DR has usually performed quite well for robust policies. Can you explain this marginal increase? "
> >> We stated in the paper that we implemented a *limited* form of domain randomization whereby we sample the same set of perturbations that we use in the robust uncertainty set to ensure a fair comparison of the two techniques. We wanted to show in Figure 3 that Robust MPO makes better use of the same set of perturbations compared to a data augmentation technique like domain randomization. We do note that when significantly increasing the number of perturbations domain randomization performance improves but does not outperform Robust MPO with *only* 3 perturbations.
>
> "Could the authors please include a qualitative evaluation of the learned controllers? Especially as the Cartpole stabilization  is a linear system one could compare against the analyitc optimal controllers."
> >> We have videos in the URL (which can be found in the abstract) which provides performance of the learned controllers for MPO vs. Robust MPO on the unseen holdout set of task perturbations. One very interesting analysis is on the Walker Walk task where we have increased the thigh length of the walker. The robust agent ‘drags’ its leg as opposed to doing a normal gait movement in order to stabilize itself due to this change in length. The normal gait movement is typical of regular controllers as well as the MPO agent. So Robust MPO is learning a fundamentally different policy in this instance to that of the non-robust agents.
>
> "Minor Comments"
> "What does the bar mean, e.g., \bar{J} & \bar{\pi} mean? I cannot find a definition of the bar."
> >> This is defined in the 3rd paragraph on page 3 where \bar{J} = E_\pi[Q^{\pi_k}].
> \bar{pi} is defined on the 3rd last paragraph of page 2 as a reference policy.
>
> "Typo in the Background section 'Kullback-Liebler  (KL)'"
> >> Thanks, we have corrected this.

---

### Decision · Program_Chairs · 2019-12-19

**Decision:**

Accept (Poster)

**Comment:**

The authors provide a framework for improving robustness (if the model of the dynamics is perturbed) into the RL methods, and provide nice experimental results, especially in the updated version. I am happy to see that the discussion for this paper went in a totally positive and constructive way which lead to a) constructive criticism of the reviewers b) significant changes in the paper c) corresponding better scores by the reviewer. Good work and obvious accept.